# Collaborative and privacy-preserving retired battery sorting for profitable direct recycling via federated machine learning

Shengyu Tao [1,6], Haizhou Liu[1,6], Chongbo Sun[1,6], Haocheng Ji[1], Guanjun Ji[1], Zhiyuan Han[1], Runhua Gao[1], Jun Ma[1], Ruifei Ma [1], Yuou Chen[1], Shiyi Fu[2], Yu Wang[2], Yaojie Sun[2], Yu Rong[3], Xuan Zhang [1] ✉, Guangmin Zhou [1] ✉ & Hongbin Sun[1,4,5] ✉

Unsorted retired batteries with varied cathode materials hinder the adoption of direct recycling due to their cathode-specific nature. The surge in retired batteries necessitates precise sorting for effective direct recycling, but challenges arise from varying operational histories, diverse manufacturers, and data privacy concerns of recycling collaborators (data owners). Here we show, from a unique dataset of 130 lithium-ion batteries spanning 5 cathode materials and 7 manufacturers, a federated machine learning approach can classify these retired batteries without relying on past operational data, safeguarding the data privacy of recycling collaborators. By utilizing the features extracted from the end-of-life charge-discharge cycle, our model exhibits 1% and 3% cathode sorting errors under homogeneous and heterogeneous battery recycling settings respectively, attributed to our innovative Wasserstein-distance voting strategy. Economically, the proposed method underscores the value of precise battery sorting for a prosperous and sustainable recycling industry. This study heralds a new paradigm of using privacy-sensitive data from diverse sources, facilitating collaborative and privacy-respecting decision-making for distributed systems.

Lithium-ion batteries (LIBs), serving as energy storage devices, have gained widespread utilization across various domains, from industry production to daily life as an accepted technical route. Projections suggest that the global production scale of LIBs will surpass 1.3 TWh by 2030[1] when the escalating demand for batteries will far outpace the availability of vital metal resources like lithium and cobalt[2,3]. However, the current average lifespan of LIB products stands at 5–8 years, leading to an imminent surge in retired batteries in many countries. If not appropriately managed, retired batteries will result in unsustainable resource wastage and environmental harm. Given these

circumstances, the development of battery recycling technology assumes crucial importance as we confront the impending tide of LIB retirements[4].

Recent advances in battery recycling research have been focused on the pyrometallurgical, hydrometallurgical, and direct recycling approaches[5]. In contrast to the pyrometallurgical and hydrometallurgical methods, direct recycling stands apart as a distinct approach. This process does not inflict secondary damage on the material structure, enabling more efficient structural repair and performance restoration. Moreover, direct recycling exhibits higher

[1]Tsinghua-Berkeley Shenzhen Institute, Tsinghua Shenzhen International Graduate School, Tsinghua University, Shenzhen, China. [2]School of Information Science and Technology, Fudan University, Shanghai, China. [3]Tencent AI Lab, Tencent, Shenzhen, China. [4]Department of Electrical Engineering, Tsinghua University, Beijing, China. [5]College of Electrical and Power Engineering, Taiyuan University of Technology, Taiyuan, China. [6]These authors contributed equally: Shengyu Tao, Haizhou Liu, Chongbo Sun. ✉e-mail: xuanzhang@sz.tsinghua.edu.cn; guangminzhou@sz.tsinghua.edu.cn; shb@tsinghua.edu.cn

profitability, as it entails lower energy consumption, reduced greenhouse gas emissions, and lighter environmental footprints[6,7]. In actual production, however, battery recyclers frequently encounter LIBs comprising unknown components or battery modules that consist of a mixture of different cathode material types. Considering that direct recycling can be heavily cathode-specific, such a complexity renders the application of direct recycling infeasible for achieving value conversion of the retired batteries[8]. It is crucial to emphasize that even if the vital metals from mixed cathode material types can be extracted using conventional recycling strategies, the interplay between different cathode materials during the recycling process can adversely impact product quality[9]. Therefore, understanding cathode material type information on the recycling side markedly impacts the direct recycling route choice and ultimately improves product quality, profitability, and sustainability.

Human-assisted direct recycling has been proposed to identify retired battery cathode material type information in the pre-treatment link, which is still not financially viable when the recycling industry is scaling up[1]. To effectively retrieve the retired battery cathode type information, the scientific and industrial community has recently initiated a battery lifetime tracing system[10] and emerging concepts like battery passport[11] and battery data genome[12]. Although substantial batteries have been utilized before those initiatives, there is a growing consensus that battery information should be accessible throughout the life chain to facilitate second-life decision-making[13]. This is notably the case for the battery recycling sector, the last station of the battery's second life, as the recycling route can be heavily cathode-specific. However, battery lifetime tracing systems or battery passports are enabled by electronic gadgets like bar codes and near-field communications, which could introduce intensive investment and could be widely incompatible with different battery designers. Furthermore, electronic gadgets remain challenging to consistently manage throughout their lifespan, leading to worn-out devices and inaccessibility at the recycling stage since the modern manufacturing process of LIBs is still not production-to-recycling integrated[14]. Hence, more breakthroughs are urgently needed to achieve an efficient battery cathode type sorting only using easy-to-access field information[15,16], opposite to the historical data recorded or the human-assisted manner, facilitating the adoption of direct recycling to improve the quality and profitability of recycled products.

In the past few years, machine learning has emerged as a viable tool to tackle open questions in all battery fields. In other battery-related topics, machine learning has recently allowed us to automatically discover complex battery mechanisms[17–19], predict remaining useful life[20–24], evaluate the state of health[19,25,26], optimize the cycling profile[27,28], approximate the failure distribution[29], even to guide the battery design[30,31], and predict life-long performance immediately after manufacturing[32]. In the case of battery recycling, few works have investigated machine learning regarding cathode materials[33,34], which blames the scarce battery data, especially for those cycled to the end-of-life stage. The vast majority of published studies showcase very limited sample sizes[35] and are even more limited in battery cathode diversity[36]. The scarcity is attributed to the intensive cost, the long testing time[37], and, most importantly, the data privacy due to commercial or interest concerns. Consequently, the privacy issue rigidifies the dilemma where the existing battery data, though substantial in volume and diversity from multiple parties such as battery manufacturers, practical applications, academic institutions, and third-party platforms, cannot be shared. Such a dilemma calls for studying the cathode material sorting to optimize battery recycling route choice in a collaborative while privacy-persevering fashion.

Federated machine learning, as a distributed and privacy-preserving paradigm, has the potential to resolve both multi-party collaboration (equivalently, the battery data volume and diversity) and privacy issues through collaborative machine learning[38–40]. In each training iteration, the distributed data owners perform local training with their local computational power, encrypt the as-trained model parameters/results, and upload them to a central coordinator for aggregation. Facts that raw datasets never leave their respective data owners and that transferred parameters/results are properly encrypted to protect data privacy. Federated machine learning has been extensively investigated in numerous applicative fields, including public health[41,42], clinical diagnosis[43–45], e-commerce[46], Internet of Things[47], mobile computing[48], and smart grid[49–51]. This approach can revolutionize the data-driven research paradigm in wide energy sectors by enabling privacy-preserving collaboration, especially for those with limited data access. Regarding the battery recycling sector, federated machine learning assumes promising possibilities for leveraging the giant amount of battery data that already exists but cannot be shared due to privacy concerns. With such a collaborative while privacy-preserving paradigm, retired battery sorting can be implemented with high accuracy, efficiency, scalability, and generalization, optimizing the quality and profitability of recycled products. To our knowledge, federated machine learning studies focused on battery recycling have never been reported.

In this study, we perform a cathode material sorting of the retired batteries, leveraging the existing battery data from multiple collaborators, such as battery manufacturers, practical application operators, academic research institutions, and third-party platforms, in a collaborative while privacy-preserving machine learning fashion as illustrated in Fig. 1. Our federated machine learning model was trained using only one cycle of field testing data via a standardized feature extraction process, without any prior knowledge of the historical operation conditions. We compare the predictive power of our federated machine learning model with that of independently learned local models based on local data under both homogeneous and heterogeneous battery recycling circumstances. The heterogeneity issue is resolved by our proposed Wasserstein-distance voting strategy. An economic evaluation of retired battery recycling using our proposed federated machine learning framework is conducted, highlighting the relevance and necessity of accurate sorting of retired batteries. We comprehensively discuss the model interpretability, battery recycling implications, and broader prospects of the future battery recycling practice integrated with federated machine learning.

## Results
### Data collection and standardization
The unique battery kinetics in different battery types are often high-dimensional and hard to characterize due to divergent operating cases, manufacturing variability, and historical usages[52]. To find a solution to this dilemma, we collected and standardized 130 retired batteries with 5 cathode material types from 7 manufacturers to construct an out-of-distribution, equivalently heterogeneous dataset. Given different historical usages, the capacities of the collected batteries are below 90% of the nominal capacity. The battery cathode materials are lithium cobalt oxide (LCO), nickel manganese cobalt (NMC), lithium ferrophosphate (LFP), nickel-cobalt-aluminum oxide (NCA), and NMC-LCO blended types, which are further grouped into 9 classes based on the manufacturers (Supplementary Table 1). We intentionally include batteries with divergent historical usages, from laboratory testing to electric vehicle driving profiles, to train a generalized model for the battery recycler independent of historical usages and battery types.

For standardization, all data required from the recycler are the currently-probed (field-testing) cycle with one charging and discharging test, which is easy to implement in practical cases. The as-probed data are first denoised by filling in missing values, replacing outliers, and performing median filtering. Human-induced and cathode-heterogeneity-induced noises are deliberately retained, though, to make the model robust to imperfect inputs. The data are then linearly

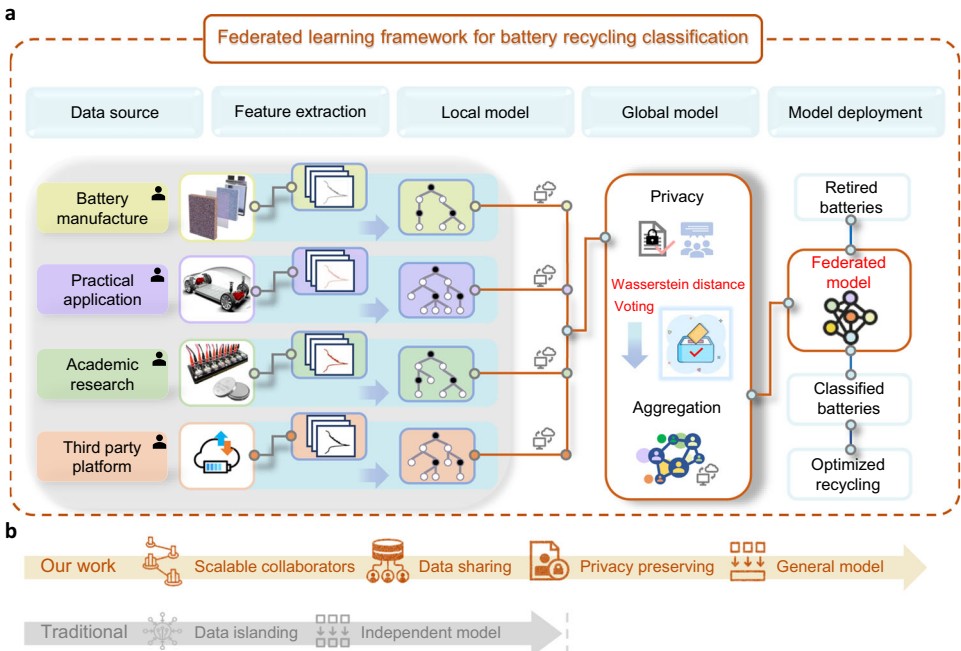

**Fig. 1 | The federated machine learning framework of retired battery sorting for recycling. a** Multiple data sources, such as battery manufacturers (Image courtesy Addionics), practical application operators (battery pack in the floor pan of a Tesla. Image courtesy of Tesla), academic research institutions, and third-party platforms, can be data contributors. The battery data are neither exchanged between contributors nor uploaded to the battery recycler. Instead, the data contributors train local models and share model parameters with the battery recycler to build a global model. The proposed Wasserstein-distance voting technique fuses the local models into the global model, which is robust to data imbalance and noise. Battery recyclers can use the jointly-built model for battery sorting, combined with the easy-to-access field testing data. **b** Our federated machine learning framework encourages collaborators to sharing the data while preserving data privacy as apposed to the traditional data islanding paradigm.

interpolated for curve filling (Supplementary Fig. 1) and feature engineered for dimensionality reduction, with a shared set of standardization parameters (Supplementary Note 1). Features extracted from the standardization pipeline are well interpretable, a concern of significant commercial interest. To the best of our knowledge, it is the first time that heterogeneous battery data from multiple sources and historical usages are utilized to assist in the strategy design of battery recycling.

Figure 2a, b demonstrate the feature engineering process. We focus on the charging and discharging curve of the retired batteries in the last cycle, i.e., one charging and one discharging cycle (Supplementary Figs. 2–5). In the charging cycle, 15 features are extracted from the voltage-capacity and dQ/dV curves, where V and Q refer to the voltage and capacity values, respectively. The same set of features are extracted for the discharging cycle. As a result, 30 features are extracted in total, as indicated from F1 to F30. Refer to Supplementary Table 2 and Supplementary Note 2 for a detailed explanation of the features. Figure 2c showcases the absolute and relative feature values of the selected batteries from each class. Most relative feature values in different classes overlap in the −1 to 0 region (with the light green color) and are indistinguishable, illustrating the difficulty in classifying battery type using one cycle of battery data. The difficulty is expected because the divergent historical operation conditions can influence the charging-discharging kinetics of the batteries so that the extracted features can be largely correlated despite the different battery types (Supplementary Fig. 6). Rather than directly interpreting the extracted features using expert knowledge, we employ an alternative data-driven approach that automatically leverages the latent patterns across various battery types.

### Retired battery sorting with homogeneous data access

We first consider a setting where the battery data are homogeneously distributed across the collaborators (namely, the clients). The homogeneity means that each client offers to share the battery data across all 9 classes, even though the specific number of batteries is not restricted (Supplementary Table 3). We train our federated machine learning model without requiring information on the historical use of the retired batteries. In our work, the recycler and the clients only need to test the retired batteries at the current (field-testing) cycle, specifically, with a complete charging-discharging cycle for a standard feature engineering process initiated by the recycler. Local models are trained based on features extracted from their private battery data. The federated machine learning framework aggregates the local model parameters, rather than the private battery data, for the recycler to classify the retired batteries.

Figure 3 shows the sorting results when clients contribute homogeneous battery data. Figure 3a compares two federated machine learning methods, i.e., the majority voting (MV) and our proposed Wasserstein distance voting (WDV), with the independent learning (IL) paradigm. It should be noted that the accuracy for the IL is averaged over all clients in a non-federated manner. Compared with the IL, the MV does not sacrifice sorting performance, with an average accuracy of 95%, while being capable of protecting data privacy and mitigating computational burden. However, 3 classes are missorted using the MV. For instance, 3 batteries in NMC (SNL, class 8, 15 in total) are missorted into NCA (SNL, class 7), resulting in a sorting accuracy of 80%. The sorting accuracy for NCA (UL-PUR, class 9) is 81%, with 2 batteries missorted into NMC (MICH_Form, class 4) and 1 battery missorted into NMC/LCO blended type (HNEI, class 2), respectively. In contrast, the WDV outperforms the MV since it only missorted one battery, resulting in a sorting accuracy of 99%. We also evaluate the prediction probability of each class for the MV and WDV, respectively. It turns out that the WDV makes a more confident sorting than the MV since the prediction probabilities of the WDV are generally right-skewed to a higher probability value. Therefore, our proposed WDV produces higher sorting accuracies across all classes, and the sorting is of richer probability confidence margins.

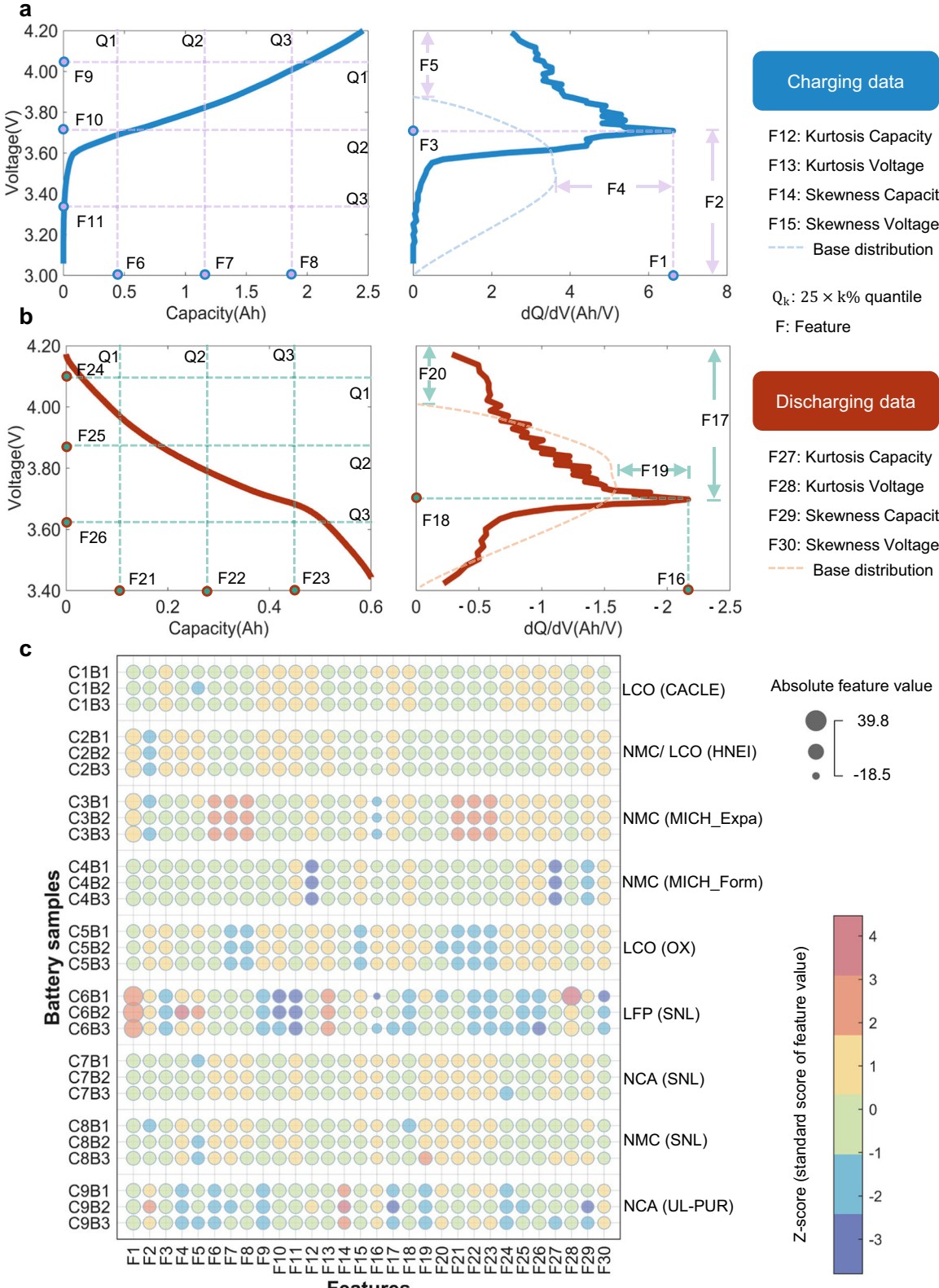

**Fig. 2 | The feature engineering result. a** For the charging process, 15 features are extracted from the voltage-capacity (left) and dQ/dV curve (right). **b** The same set of features are for the discharging process as F16 to F30. **c** Features are visualized by classes, following the format CxBn, indicating the nth battery from class x. The size of a circle maps the absolute feature value. Source data are provided as a Source Data file.

We also evaluate the privacy budget (PB, "Methods" section), considering that client data might be vulnerable to reverse engineering by eavesdropping on private data[53]. In this regard, we add random Gaussian noise to the client data with different intensities. The intensity of the randomness is controlled by a noise-to-signal ratio (NSR), ranging from 1% to 10%. Figure 3b shows the accuracy and privacy budget comparison when using IL, MV, and WDV, respectively. The sorting accuracy of the MV decreases from 95% to 82%, similar to

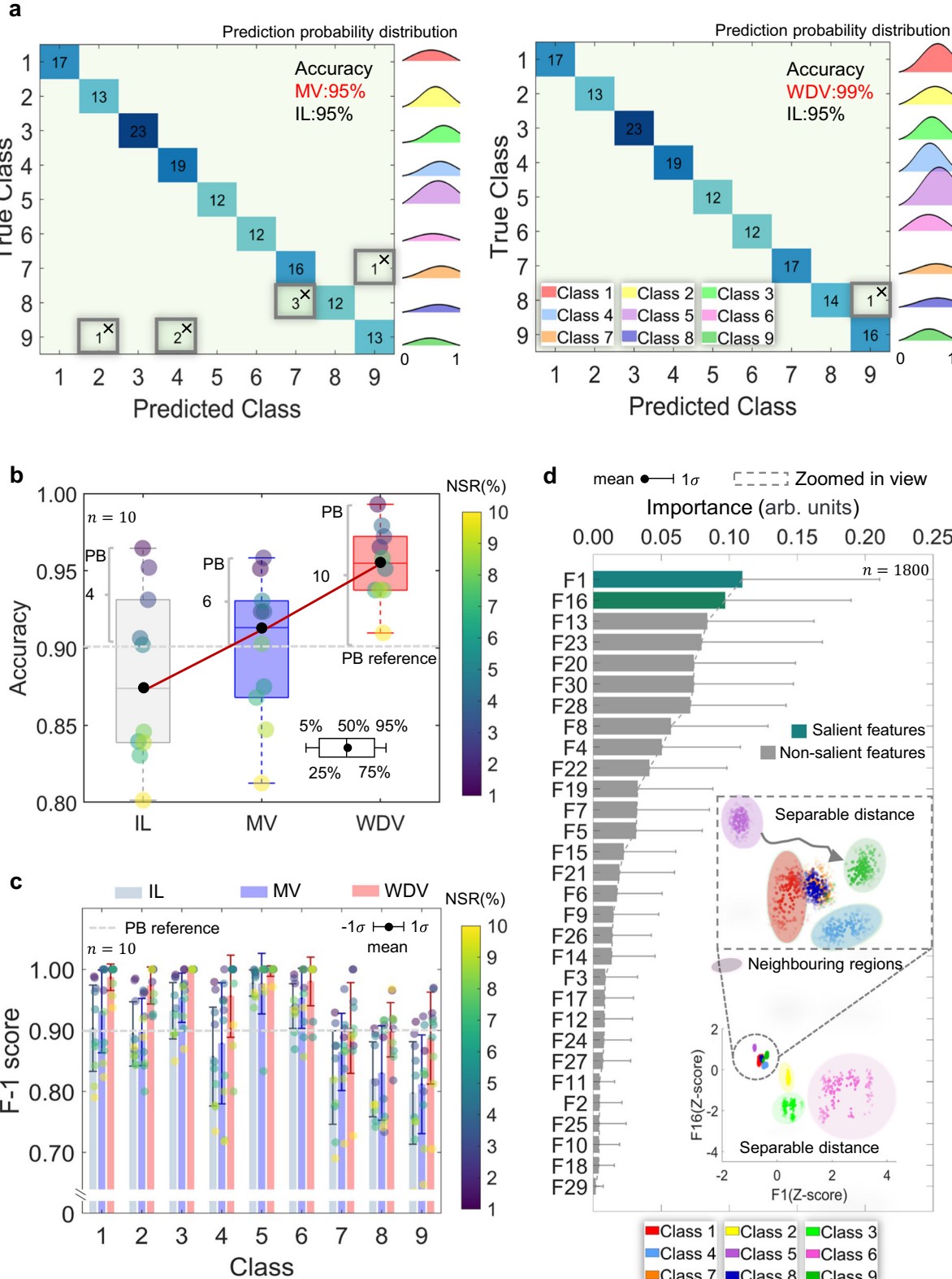

**Fig. 3 | Sorting results when clients have homogeneous data access. a** The confusion matrix for the majority voting (MV) and Wasserstein distance voting (WDV) methods, respectively. We consider the prediction probability distribution for each class. The sorting of independent learning (IL) is annotated. **b** Sorting accuracy distribution and privacy budget (PB) of the IL, MV, and WDV in the presence of random noise. The PB value is referenced at a 90% accuracy level.

**c** Average F1-score of sorting results and PBs in each class using the IL, MV, and WDV. The PB values are all referenced at a 0.9 F1-score level. Data are presented as mean values ±1 standard deviation. **d** Feature importance, in descending order. The subplot shows the feature space spanned by the first two most salient features. Data are presented as mean values + 1 standard deviation. Source data are provided as a Source Data file.

that of the IL when the noise intensity increases from 1% to 10%. In this noise range, the median sorting accuracy of the MV and IL is 92% and 86%, respectively. In contrast, the sorting accuracy of the WDV is still above 90% in the presence of 10% noise, which is a stringent noise level in practical cases. The WDV has a median sorting accuracy of over 95% in the same noise range. Taking an average sorting accuracy of 90% as an acceptable reference level, the PB values of IL, MV, and WDV are 4, 6, and 10, respectively. Therefore, applying federated machine learning produces a more privacy-secure sorting than IL, hence being capable of preventing data eavesdropping. Furthermore, our proposed WDV is more accurate and performs much better (with nearly doubled PB values) in the privacy-accuracy trade-off than the MV. In addition, the robustness to stringent noise when using the WDV also implies a good tolerance of the battery measurement requirement, reducing the expensive battery testing disbursement.

Noticing that a high sorting accuracy does not necessarily imply an acceptable sorting for a specific class, we also consider within-class sorting performances. Figure 3c shows the F1-score and privacy budget of the IL, MV, and WDV in each predicted class; note that the privacy setting is identical to that in Fig. 3b. The result shows that the IL has smaller F1-scores than the federated machine learning manner in all the classes, making poor sortings. Regarding federated machine learning, WDV outperforms the MV in each predicted class by producing higher average F1 scores. The deviation range of the F1 scores for WDV is smaller than that of the MV, indicating that the WDV is more robust (Supplementary Fig. 7). Therefore, our proposed WDV not only has a better overall sorting accuracy among all nine classes (Fig. 3b) but also within each class, compared with the MV. Regarding the privacy budget, the PB value when using the non-federated IL, referenced at a 0.9 F1-score level, is significantly lower than the federated way (Supplementary Table 4) across all classes. This indicates a more severe risk of data leakage for IL compared with federated machine learning. When further applying our proposed WDV, the private budget can increase by 78% and 44% compared with the non-federated IL and the federated MV, respectively (Supplementary Table 4). The results demonstrate that the WDV successfully leverages the battery-chemistry-related insights hidden in clients while effectively preserving client data privacy.

We then interpret our federated machine learning model by evaluating the most salient features correlated with battery cathode chemistry. Figure 3d shows the importance of the features in descending order. The error bar indicates the importance deviation. Features F1 and F16 rank the top two features regarding out-of-bag importance ("Methods" section). Interestingly, these two features have a clear physical interpretation of the battery dynamics, which we will further discuss in later sections. Here, we rationalize these two features by plotting the grouped battery samples in the feature space spanned by features F1 and F16. The subplot of Fig. 3d shows that NMC/LCO blended type (HNEI, class 2), NMC (MICH_Expa, class 3), and LFP (SNL, class 6) (sharing the color with Fig. 3a) are clearly separable in the spanned feature space. For the remaining classes, the batteries are still separable (see the zoomed-in view), though in relatively more minor grains. On the contrary, the non-salient features have a relatively weaker sorting ability due to the non-separable feature space spanned (Supplementary Fig. 8). As a result, our federated machine learning framework successfully discovered useful mechanism insights to guarantee sorting accuracies. Such an insight could be further extended to simplify the model for light computation, hence less investment. Once the client models classify the batteries, the recycler can aggregate the client results to make a final decision on the battery cathode material types underpinned by the salient features.

## Retired battery sorting with heterogeneous data access

We also consider an extreme, while a more actual situation where the data can be exclusively scattered among clients, i.e., the data distribution is heterogeneous. In this situation, the heterogeneity issue poses more challenges to battery type sorting since the clients are prone to train biased models and deteriorate global accuracy, which is still an open question in federated machine learning. In this section, we explore a more challenging situation rather than having homogeneous data access among each client (Supplementary Note 3). We demonstrate that our federated machine learning framework can still classify retired batteries based on the standard feature engineering process at the current (field-testing) cycle without any knowledge of the previous operation conditions.

Figure 4 shows the sorting results when clients have heterogeneous data access. We consider the heterogeneity index, defined as the minimum number of battery classes for each client in each Monte Carlo simulation run. A higher heterogeneity index indicates a less heterogeneous battery data distribution. The heterogeneity index is no smaller than two such that one client can train a local model for a sorting task. Figure 4a shows average sorting accuracy when the heterogeneity index varies. The average accuracies are plotted with solid lines, with the ±1 standard deviation range indicated in the shaded region. As the heterogeneity index decreases from 9 to 2, the performance of the MV and the IL rapidly deteriorates at a sublinear rate. The average sorting accuracy of the MV is 0.55, slightly better than the IL, equivalent to a random guess when the heterogeneity level is two. This observation shows that the MV can help little to aggregate the local models under heterogeneous data access. In contrast, the WDV outperforms its MV counterpart in all heterogeneity levels, successfully mitigating the heterogeneous data distribution issue. Moreover, the WDV shows an interesting asymptotic effect when the heterogeneity index increases. This indicates that the WDV can potentially support the optimal allocation/distribution of client battery data to reduce the collaboration cost in practical battery recycling situations.

We select the best model using the MV when the heterogeneity index equals two and compare it with the sorting result of our proposed WDV under the same setting. The selected best model has an average sorting accuracy of 71%, as shown in Fig. 4a.

The detailed battery data distribution setting of the best model using MV is illustrated in Fig. 4b, which is heterogeneous (Supplementary Table 5). For instance, client 2 contributes to all battery classes except for NMC (MICH-Expa, class 3), while client 5 only contributes to NMC/ LCO blended type (HNEI, class 2) and NMC (SNL, class 8). Under the heterogeneous data distribution setting in Supplementary Table 5, we further compare the class-wise and client-wise sorting performance of the MV and the WDV to the non-federated IL with two considerations: (1) the significance of our federated machine learning framework and (2) why our proposed WDV outperforms the MV. First, we evaluate the client-wise sorting accuracy, shown in the lower side of Fig. 4c. Client 5 achieves an average sorting accuracy of 25%, ranking last among all clients. Meanwhile, client 2 achieves an average sorting accuracy of 86%, ranking first among all clients. However, the average sorting accuracy is only 55%, close to a random guess. Therefore, the client performance using the non-federated IL depends heavily on data access (Supplementary Fig. 9). In fact, without our federated machine learning framework, the battery recycler is equivalent to a single client, and the battery recycler can only make sortings on the battery types stored in its local database. This non-federated paradigm could not handle various types of retired batteries if the recycler did not build a database covering all the battery types it would handle. With our federated machine learning framework, the recycler can collaborate with several clients, even if under heterogeneous data situations.

We turn to analyze how to collaborate with clients under heterogeneous data access settings. The upper part of Fig. 4c shows the class-wise accuracy of the MV and WDV. It is noticed that the average sorting accuracy after using the MV is better than the non-federated way, which is 79%, as indicated in the lower side of Fig. 4c. It demonstrates the success of applying the federated machine learning

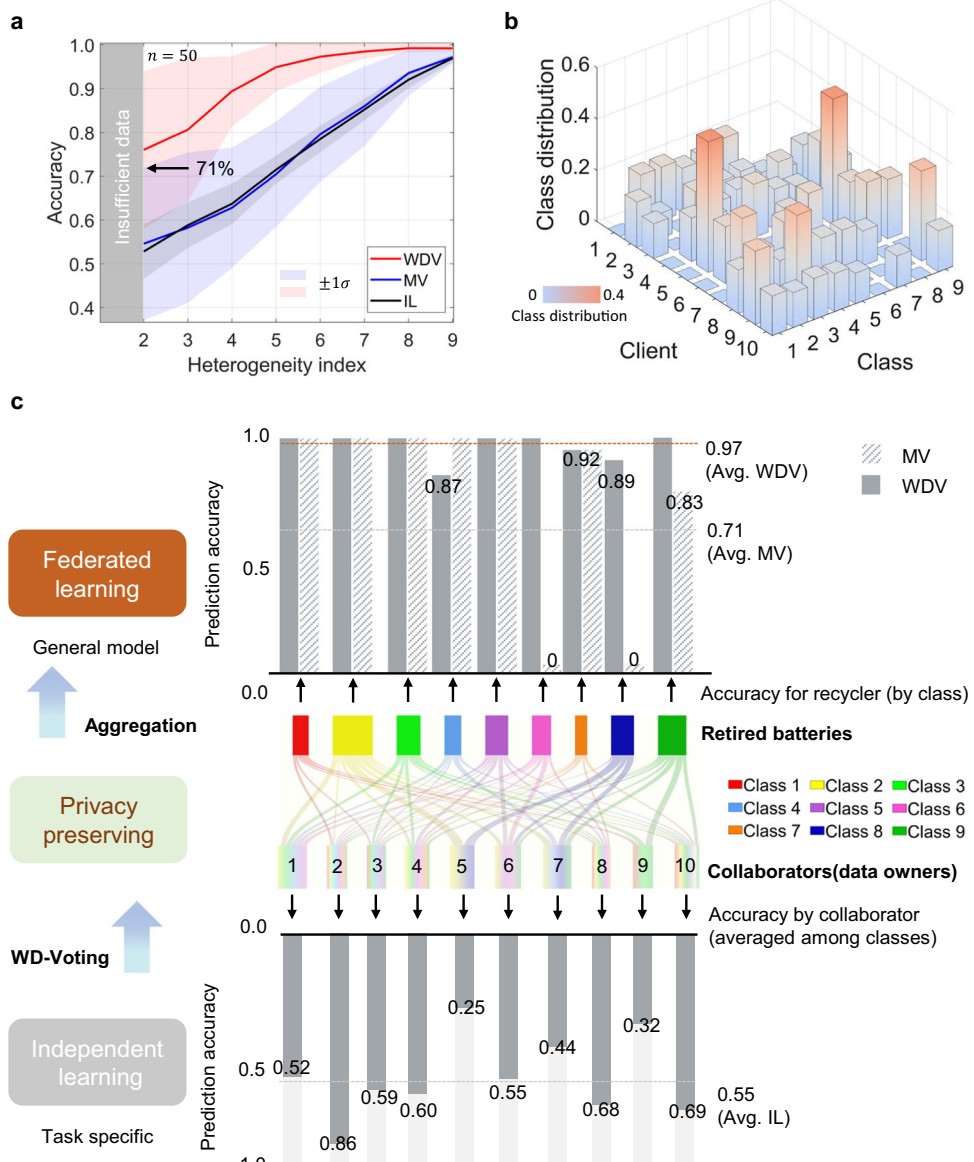

**Fig. 4 | Sorting results when clients have heterogeneous data access. a** Sorting accuracy as a function of heterogeneity index. The results are averaged over 50 Monte Carlo runs ($n = 50$), with one standard deviation region ($\pm 1\sigma$) indicated by shaded color. **b** The data distribution when benchmarking the best majority voting (MV) performance. **c** Class-wise (upper part) and client-wise (lower part) sorting accuracy corresponds to our federated and independent machine learning (IL) methods. The Sanky chart (middle) presents the heterogeneous data distribution among clients. Source data are provided as a Source Data file.

framework to address the heterogeneous data distribution issue in this case. However, the MV totally missorted LFP (SNL, class 6) and NMC (SNL, class 8) with zero accuracy. The failure of the MV in specific classes can be rationalized by its core idea of giving more weight to the clients who contribute more battery samples while not guaranteeing diversity in battery types. For instance, the contribution of client 7 will be strengthened by the MV due to a large number of batteries (specifically, 195 augmented batteries, ranking second among clients), despite only contributing four classes of batteries. As a result, the MV will lead the aggregated model to be biased towards large client such as client 7 (Supplementary Table 4). The biased phenomenon is evidenced by the as-described zero sorting accuracy for LFP (SNL, class 6) since the large client, such as client 7, never contributed any batteries in class 6. Similarly, client 1, the largest client with 197 augmented batteries, failed to contribute helpful information to the recycler regarding classifying NMC (SNL, class 8), which is consistent with zero accuracy in class 8. In contrast to the MV, our proposed WDV focuses

on the battery similarities between the recycler and each client by measuring the pairwise distance. We aim to assign fewer weightings to the clients with biased data distributions (equivalently, higher heterogeneity), whose batteries are of higher similarities with the recycler, such that the recycler can have generalized information from each client. The results show that our proposed WDV successfully leverages helpful information from heterogeneous data distribution among clients. The WDV achieves 100% and 89% sorting accuracy for the otherwise missorted batteries in LFP (SNL, class 6) and NMC (SNL, class 8), respectively. The overall sorting accuracy using the WDV is up to 97%, with only 5 batteries missorted out of 144 samples. In Supplementary Fig. 10, we also notice that the missorted batteries are of similar cathode materials. Specifically, 2 batteries with the NMC cathode material were missorted into the NMC/LCO blended type; while 1 battery with the NCA cathode material was correct in material type but missorted into another manufacturer. On the contrary, the missorted results produced by the MV can spread to either many irrelevant

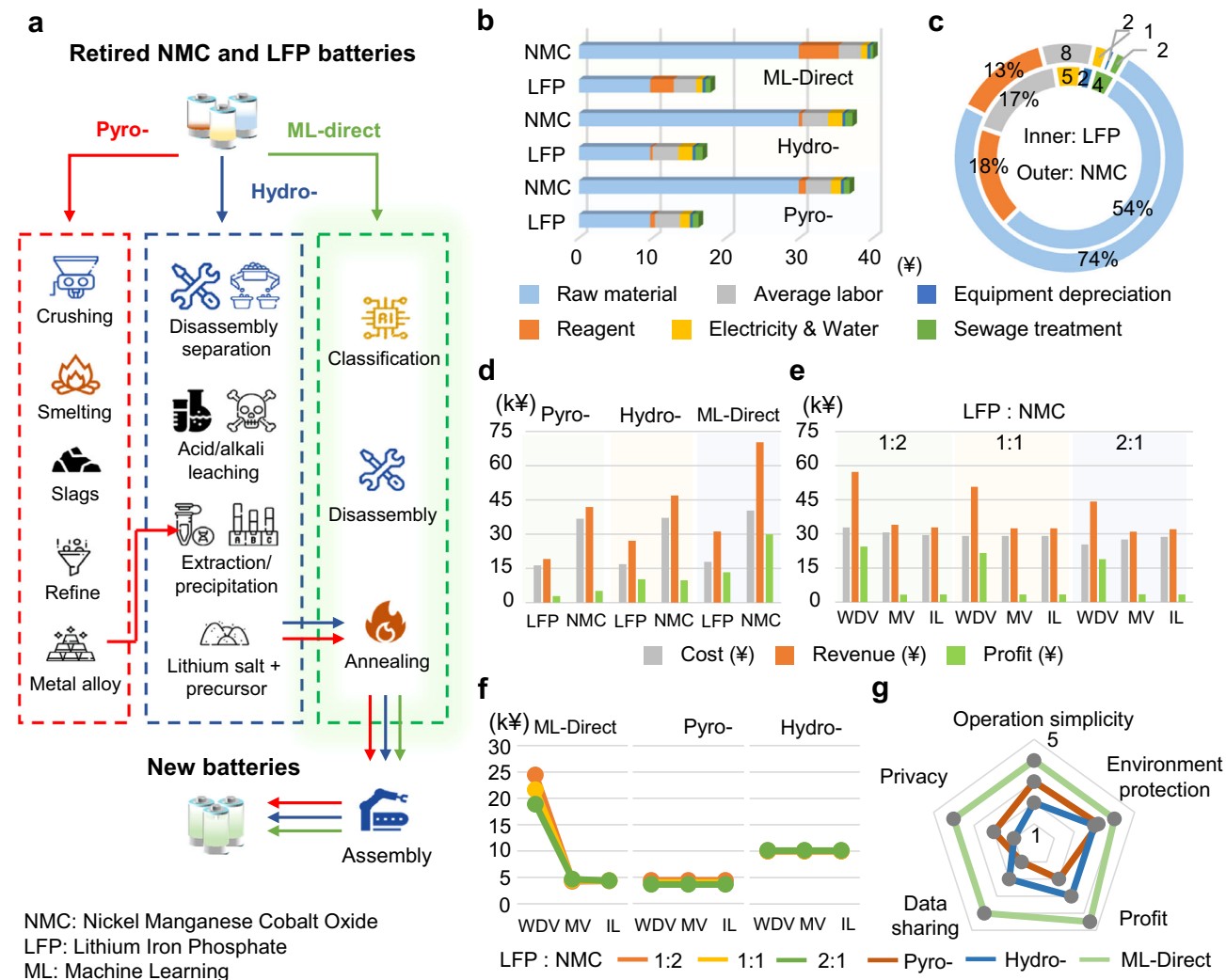

**Fig. 5 | An economic evaluation of retired battery recycling. a** Comparison of the Pyro- (pyrometallurgical), Hydro-(hydrometallurgical), and ML-direct (machine learning aided direct) recycling methods. **b** Cost analysis of Lithium Iron Phosphate (LFP) and Nickel Manganese Cobalt Oxide (NMC) batteries using different recycling methods in individual modes. **c** Cost analysis of LFP and NMC batteries using ML-direct recycling in individual mode. **d** Cost, revenue, and profit comparison of the individual battery type using different recycling methods in individual mode. **e** Cost, revenue, and profit comparison using Wasserstein distance voting (WDV), majority voting (MV), and independent learning (IL) methods in hybrid mode. The ratio is the amount of LFP battery to that of NMC battery. **f** Sensitivity analysis of the profit of WDV, MV, and IL methods in a hybrid model towards sorting accuracy in hybrid mode. The ratio is the amount of LFP to that of the NMC battery.
**g** Comprehensive comparison of different battery recycling technologies in hybrid mode. Source data are provided as a Source Data file. The graphics in panel a were created using icons from Flaticon.com.

classes or manufacturers. Therefore, we conclude that the WDV can aggregate helpful client insights by distinguishing inherited differences in cathode material types. Inspired by this, the WDV also suggests that the clients are encouraged to contribute more battery data in diversity rather than more data in some specific classes. The recycler can optimize the benefit distribution based on helpful client information provided. Ultimately, our federated machine learning framework enables the recycler to know the battery cathode material type, even if without their own data access to various battery data, while preserving the data privacy of potential clients.

### An economic evaluation of retired battery recycling

To help understand the relevance and necessity of battery sorting in actual recycling practice, also to verify the significance of our proposed WDV strategy, an economic evaluation is performed. Three recycling methods (pyrometallurgy, hydrometallurgy, and direct recycling), two battery cathode types (LFP-graphite and NMC-graphite), two recycling modes (individual, hybrid), three sorting

accuracy levels (97%, 71%, 55%) induced by the federated and non-federated machine learning methods (WDV, MV, IL) are included in the evaluation. The notation of ML-direct in Fig. 5a denotes direct recycling enabled by our federated machine learning framework. The individual mode denotes that batteries have been previously sorted in a human-aided manner (Fig. 5b–d), which is used to compare different recycling methods given a known cathode type. The hybrid mode denotes that batteries are collected with mixed cathode types (Fig. 5e–g), which is used to analyze the significance of the battery sorting toward recycling profits. The detailed calculation procedure and numerical results can be found in Supplementary Note 4 and Supplementary Tables 6–15, respectively.

Figure 5a shows a schematic diagram of three recycling methods, including pyrometallurgy, hydrometallurgy, and ML-direct recycling. The final product of pyrometallurgy is metal alloy. While final products of hydrometallurgy are lithium salt and precursor, which should be further processed to assemble batteries, as indicated by red and blue arrows in Fig. 5a. Compared to the other two non-machine

learning-aided methods, ML-direct recycling has the shortest process flow since the product is standard battery materials, which brings about the largest possible convenience and the least possible environmental footprints. It should be stressed that such convenience is enabled by accurate sorting, a vital link in pretreatment for actual battery recycling practice, thanks to our federated machine learning framework.

The cost analysis of LFP and NMC batteries using different recycling methods is shown in Fig. 5b, including raw material, reagent, average labor, electricity & water, equipment depreciation, and sewage treatment. It can be observed that the raw material accounts for the largest proportion of the cost. As a result, the cost of NMC is always higher than LFP in any method owing to the large price difference between NMC and LFP. Besides, for the same type of batteries, the cost of ML-direct recycling is the largest while the pyrometallurgy is the least, owing to the large expense of reagents. Considering the reagents can be heavily cathode material specific, the profitability of ML-direct recycling largely depends on the sorting accuracy of the mixed retired batteries. Further analysis of the detailed proportion of cost structure in ML-direct recycling is summarized in Fig. 5c. The outer and inner annuluses stand for NMC and LFP batteries, respectively. Except for raw material and reagent, the sum of the other costs is the same in price (5620 ¥/t) but more than twice the difference in percentage (NMC for 28%, LFP for 13%). The cost of raw materials for NMC (29900 ¥/t, accounting for 74%) is nearly three times that of LFP (9687.5 ¥/t, accounting for 54%), which again indicates the profit of ML-direct recycling is sensitive to the sorting accuracies. Figure 5d lists the cost, revenue, and profit of LFP and NMC batteries using different recycling methods. For the largest profit option, NMC battery using ML-direct recycling (29944.25 ¥/t) is 2.25 times the second largest profit option (LFP batteries using ML-direct recycling, 13279.51 ¥/t). It can be summarized that ML-direct recycling has the largest revenue and profit. Moreover, it is also noticed that the profit of recycling NMC is always larger than LFP, highlighting the significance of efficiently sorting high-value recycling candidates from a bulk of mixed retired batteries.

In a practical scenario, collected retired batteries could be expensive and even impossible to sort by human-aided pretreatment, especially when the recycling is scaling up. On the contrary, ML-direct recycling has the unique advantage of efficiently sorting the retired batteries by leveraging existing data sources from multiple battery recycling collaborators. An economic analysis using different machine learning paradigms (independent learning, i.e., IL; and federated machine learning, i.e., MV and WDV) is carried out in Fig. 5e, f. Due to the high sorting accuracy of WDV, the two types of batteries (LFP and NMC) can be completely sorted and the final product can be utilized to assemble new batteries directly. On the contrary, the MV and IL would produce significant errors in distinguishing cathode materials, thus leading to low-value products (impure materials) that are unable to be directly utilized, requiring further refining. As a result, the profit decreases asymptotically for MV and IL methods when sorting accuracy is lower than WDV, specifically 97%. NMC battery recycling using WDV-based ML-direct recycling has a high profit of 24389.33, 21611.88, and 18834.42 ¥/t for the LFP/NMC ratio of 1:2, 1:1 and 2:1, respectively, which are higher than those of pyrometallurgy (4372.32, 3994.46, and 3616.61 ¥/t) and hydrometallurgy (9957.45, 10039.27, and 10121.09 ¥/t). The profits of pyrometallurgy and hydrometallurgy are not sensitive to sorting accuracy since these methods do not require stringent retired battery cathode material information. Such a high profit from ML-direct recycling not merely stems from the inherited advantage of direct recycling but is enabled by our effective and accurate retired battery sorting. Finally, a qualitative comparison of different battery recycling technologies is illustrated in Fig. 5g. ML-direct recycling performs noticeable advantages in environmental protection[7,54], operation simplicity, privacy, data sharing, and profit. Our ML-direct recycling method has huge socioeconomic values and can quickly accelerate the development of the battery recycling industry, especially when next-generation batteries are even more complex in cathode material diversities.

## Discussion

We have successfully demonstrated our federated machine learning framework, especially our proposed WDV strategy, serving as a key to profitable battery recycling from a practical perspective. Such success is achieved by leveraging existing data sources to train a general data-driven battery sorting model, rather than an expensive human-aided sorting. Our model features a collaborative while privacy-preserving fashion, enabling the direct recycling methodology, which is currently heavily cathode-specific and sensitive to the recycling candidates. We discuss the merit of our work from a multi-level perspective, including the fundamental mechanism of battery sorting, the implication of profitable recycling, and the advantage of the federated battery recycling paradigm.

To realize the sorting of retired batteries, we extracted 30 features based on the battery charging-discharging and dQ/dV curves in the feature engineering process. In the previous sections, we have rationalized the salient features, i.e., F1 and F16, from the machine learning perspective that the feature space spanned by F1 and F16 is separable for different cathode material types, as shown in Fig. 3d. Here we aim to rationalize the physical interpretation of these two salient features. Features F1 and F16 are extracted from the dQ/dV curve, commonly used to analyze phase reactions in electrochemistry, though agnostic to the underlying mechanism. Regarding battery thermodynamics, the number of dQ/dV peaks and the corresponding voltage values can be used to analyze the reaction on electrodes and to judge the composition of the cathode material. Regarding battery kinetics, the shape of the dQ/dV curve can help analyze the transport capacity of electrons and ions inside the battery, from which the chemical properties of battery materials can be deduced. Here, the Gibbs phase law can further help the rationalization: $F = C - P + n$, where F represents the degree of freedom, C represents the number of independent components, P represents the number of phase states, and n represents external factors. When studying electrode materials, constant temperature and pressure are assumed; thus, $n = 0$. The number of independent components on the cathode is $C = 2$. Since the discharging process of LFP is a phase change process, there are two phases, i.e., $P = 2$. Since LCO, NCM, and NCA are solid solutions, only one phase exists during the discharging process i.e., $P = 1$. Therefore, the degree of freedom of LFP ($F = 0$) is lower than that of LCO, NCM, and NCA ($F = 1$). As a result, the voltage of LFP does exhibit significant change during the reaction process, consequently, there is a noticeable peak on the dQ/dV curve. On the comparison, during the charging and discharging process of LCO, NCM, and NCA ($F = 1$), the slope of the voltage change in the plateau is more significant than that of LFP, which can be reflected on the dQ/dV curve accordingly. Although LCO, NMC, and NCA ($F = 1$) have similar structures, their components, and Li-ion mobility during the charging-discharging process differ, resulting in different dQ/dV peak values, which can be interpreted from F1 and F16. While other features are possible to decipher battery kinetics, they demonstrate less importance since more complicated expert knowledge is required for further processing. However, we highlight the power of our machine learning model by automatically utilizing the information provided by F1 and F16, which have a clear physical interpretation and underpin a general and high-accuracy model. Such good accuracies are independent of historical usages and use only one cycle of end-of-life charging and discharging data. Consequently, the battery recycling collaborators realize good sorting accuracies with our proposed salient features aided by machine learning.

When the recycling collaborators successfully sort the retired batteries from the recycler, a voting procedure is performed. Noting that the data volume and data diversity of each recycling collaborator

may differ, the voting results could be biased to the specific cathode depending on the data distribution. This still hinders the profitability of battery recycling given low sorting accuracies, highlighting the significance of our WDV strategy. The result shows using our WDV-based federated machine learning framework, the battery recycling industry has a high possibility of transforming from the current human-aided battery sorting to an automatic, collaborative, and privacy-preserving fashion, with high sorting accuracy. Such effective retired battery sorting serves as a key to the battery recycling practice using direct recycling, equivalently, our ML-direct recycling. Without our method, the economic benefits of direct recycling could be greatly reduced to a level lower than traditional pyrometallurgy and hydro-metallurgy even with small errors in sorting accuracy. In next-generation battery recycling, there could be various battery types involving different chemical compositions, including Si anode, Na ion, lithium-sulfur, and zinc-air batteries, etc. The data collection for model training will be even more challenging due to data privacy and data heterogeneity (battery diversity), calling for federated machine learning to address such issues. In addition to battery type information, direct recycling also requires sorting for the state of health (SOH) since SOH directly determines the amount of reagents added to the direct recycling process, accounting for the majority of the recycling cost. Excessive or insufficient reagents will lead to a declined product quality, which in turn leads to a decline in revenue and sustainability. Different from the sorting problem, SOH estimation is more challenging since it requires historical data to formulate a regression problem. Moreover, the SOH can be heavily dependent on historical usage, while such information is difficult to retrieve at the end-of-life stage due to poor lifecycle management. Using only field-available information to determine SOH remains a critical challenge. It should be noted that the profit calculation assumes an 80% SOH for the retired batteries. Future work should consider SOH information to increase the profitability of ML-direct recycling, which is a great commercial concern.

As mentioned above, the core idea of adopting federated machine learning into battery recycling is leveraging the existing data information in a collaborative while privacy-preserving manner, which is intuitively consistent with the distributed nature of battery data. We note that the cost of sorting through machine learning is not considered, which attributes to a lack of relevant industry data and conversion standards. Under a federated machine learning setting, the recycler only needs to process battery information, thus the machine learning cost is not sensitive to the recycling scale. We, therefore, assume that the once-for-all machine learning cost will be covered when the battery recycling scale enlarges. Even though more in-depth investigations on the accuracy-privacy-cost balance should be conducted, we emphasize that the proposed federated machine learning framework tackles the common concerns in collaborative learning, including privacy, efficiency, and fairness, which can be addressed consistently and elegantly. We begin by noticing that the data privacy of the collaborating clients is fully protected, as neither the raw battery data nor the extracted features are leaked out of their respective data sources; even the as-trained local models are kept confidential to the data sources themselves. The only information being transferred, the local battery cathode sorting result, can be appropriately encrypted before transferal to eliminate potential eavesdroppers, ensuring privacy budget. Also, with the full support of parallelized local training and only one round of result transfer, the proposed framework is highly efficient in computation and communication, which remains a huge challenge in commercialization in other fields. Specifically, the selection of random forests as the bottom-level machine learning algorithm, instead of more advanced neural network architectures, is made with full consideration of the feature engineering settings and cost-effectiveness requirements. Feature engineering, which prepares the data for federated machine learning with expert-knowledge-based information extraction, transforms the raw gigabyte-scale sequential

data into kilobyte-scale tabular data. Decision trees such as random forests are more adept at learning from such low-dimensional data with heterogeneous features, whether in terms of accuracy, efficiency, or interpretability. Also, advanced neural network architectures such as Convolutional Neural Networks (CNNs) require much higher computational power from every collaborating manufacturer, with a significantly lengthened training time and compromised model interpretability, despite gaining a slight edge in accuracy (See Supplementary Fig. 11). Thus, our proposed framework is light and scalable without requiring intensive investment in the battery recycling sector, which is of significant interest to industrial practice. The framework further achieves fairness by assigning a local training task of the same scale, though in different cathode types and sample sizes, to all clients under the recycling collaboration. Even when compared with other alternative federated machine learning frameworks, the proposed framework is still better in terms of these metrics, as those alternative frameworks would most likely require various rounds of model updates with considerable parameter transferals, which would compromise efficiency and expose collaborators to an immense level of privacy leakage. Currently, the framework is implemented under the ideal assumption that all collaborators are fully cooperative, such that the uploaded local results are assumed to be reliable. Supplementary Figure 12 shows that, despite such an ideal assumption, the random forest model, incorporated with the Wasserstein distance voting, is naturally robust against random parameter transfer losses even if parameters from a few collaborators end up missing. The sorting accuracy only slightly degrades given the same heterogeneity setting. In the case of a blockchain-like environment with numerous collaborators of unknown trustworthiness and reliabilities, our research could be further extended in search of a proper incentivization mechanism such that all recycling collaborators would fully contribute to their respective local model instead of attempting to become total free-riders. We prospect quantifying the helpful information that the recycling collaborators provided to design a benefit distribution strategy and a free-rider detection scheme to make the federated battery recycling ecosystem economically feasible.

By exploring federated machine learning in the battery recycling sector, the major concern on the profitability of recycled products can be guaranteed. Our work highlights a general retired battery sorting model only using one cycle of end-of-life battery data, enabling the rational design of a direct recycling route for higher product quality and profitability in practice. The privacy-preserving information-sharing mechanism encourages extensive multi-party collaboration in battery recycling practices, thanks to our proposed WDV strategy. Our work enlightens using machine learning to facilitate an efficient and profitable next-generation battery recycling industry in the future.

To conclude, federated machine learning is a promising route for retired battery sorting and enables emerging battery recycling technologies, especially direct recycling, in their development, practical application, and optimization. We create a retired battery sorting model using only one cycle of end-of-life charging and discharging data as opposed to any historical data while preserving the data privacy budgets of multiple battery recycling collaborators. In the homogeneous setting, we obtain a 1% cathode material sorting error; in the heterogeneous setting, we obtain a 3% cathode material sorting error, thanks to our Wasserstein-distance voting strategy. Such a level of accuracy is achieved by (1) automatically exploring the unique patterns in the salient features without assuming any prior knowledge of historical operation conditions and (2) using our proposed Wasserstein-distance voting strategy to correct heterogeneous data distribution among recycling collaborators. An economic evaluation showcases the relevance and necessity of accurate retired battery sorting to the profitable battery recycling industry using direct recycling. In general, our approach can complement the existing first-principle-based recycling route research paradigms on actual battery recycling practice,

where retired batteries are necessary while challenging to sort. Broadly speaking, our work enlightens the possibilities of leveraging existing data from multiple data owners, rather than extra time-consuming and expensive data generations, to develop and optimize complex decision-making procedures such as the battery recycling route design in a collaborative while privacy-preserving fashion.

## Methods

### Privacy-informed data augmentation

We perform data augmentation with two considerations: (1) more diversified data for training a generalized model and (2) protecting data privacy by preventing adversarial reconstruction (reverse engineering to eavesdrop on the private data). Given a feature matrix $\mathcal{F}_{N \times M}$ and a class label vector $C_{N \times 1} = \{c^i\}$, $i = \{1,2,\cdots,9\}$, where $N$ and $M$ are the numbers of measured batteries (known as "observations") and associated features, respectively. The data augmentation includes the following three steps: First, we index the feature matrix $\mathcal{F}$ for a subset $F^i$ using each unique class label $c^i$. Second, we augment $F^i$ into $F^i_{\text{Aug}}$ by resampling with replacements using Bootstrapping. The resampling size (or the number of bootstrapped observations) of $F^i_{\text{Aug}}$ is set to $S = 200$. For the class label of $F^i_{\text{Aug}}$, we have $\{\tilde{c}^i_{\text{Aug}}\} = \{c^i\}$, which means the augmented class labels $\{\tilde{c}^i_{\text{Aug}}\}$ is identical to the original class labels. Third, we add random Gaussian noise to each observation of $F^i_{\text{Aug}}$ to evaluate the robustness and the privacy budget of the trained model, then we have a noisy feature matrix subset $\tilde{F}^i_{\text{Aug}}$. A hyperparameter NSR, i.e., the noise-to-signal ratio in percentage, controls the noise intensity, defined as the ratio of noise power to signal power. Intuitively, the model performance $A$ can be deteriorated by increasing NSR, denoted by a function $A(\text{NSR})$. The definition of the privacy budget (PB) is given:

$$PB = 100\% \times \ \max(\text{NSR}|A(\text{NSR}) \geq \underline{A}) \tag{1}$$

where $\underline{A}$ denotes the lower bound of acceptable accuracy of the model, depending on specific application requirements.

Finally, we stack the augmented data of each class to get the augmented feature matrix $\tilde{\mathcal{F}}_{9S \times M} = \{\tilde{F}^i_{\text{Aug}}\}$ and the class label vector $\tilde{C}_{9S \times 1} = \{\tilde{c}^i_{\text{Aug}}\}$, where $i = \{1,2,\cdots,9\}$. We use 80% (the primary split) of the augmented data to generate the client samples as the training set. We use 40% (the secondary split) of the remaining augmented data as the testing set. Both primary and secondary splits are in a stratified manner to ensure samples required in each client are sampled. The detailed data split setting here is for illustration and can be modified to further investigate the minimum data sample requirement for collaborators.

### Client Simulation

The federated machine learning framework involves multiple collaborators, known as clients, to train a global model jointly. One client serves as a data contributor for the battery type sorting task in our setting. In this work, we simulate 10 clients, each possessing different classes and different observations of battery data. To be specific, each $\text{Client}_k$ is defined over a triplet, i.e., $\text{Client}_k \triangleq (\text{lb}_k, \text{ub}_k, \text{NC}_k)$, $k = \{1,2,\cdots,10\}$, where $\text{lb}_k$ and $\text{ub}_k$ are the minimum and maximum number of observations in $\text{Client}_k$. The value of $\text{lb}_k$ and $\text{ub}_k$ are set to 100 and 200 for all clients, respectively. $\text{NC}_k$ stands for the minimum number of classes in each $\text{Client}_k$, quantifying the level of client-wise heterogeneity (namely, the heterogeneity index in the main manuscript). Then, random observations are subsequentially drawn from the augmented data $\{\tilde{\mathcal{F}}, \tilde{C}\}$ for the client based on the as-defined triplet.

### Client model

The random forest is a decision-tree-based machine-learning algorithm, with each tree defined over a collection of random variables. Formally, for an $m$-dimensional feature vector $\mathbf{X} \triangleq [x_1, \cdots, x_m]^T = \tilde{\mathcal{F}}$,

and a response vector $\mathbf{Y} \triangleq \tilde{C}$, the goal is to learn a prediction function $g(\mathbf{X})$ for predicting $\mathbf{Y}$. The prediction function $g(\mathbf{X})$ is determined by minimizing the expectation of the loss function $L$:

$$E_{XY}[L(\mathbf{Y}, g(\mathbf{X}))] \tag{2}$$

where, the subscripts denote expectations on the joint distribution of $\mathbf{X}$ and $\mathbf{Y}$.

The $j$-th decision tree, or the $j$-th base learner, is denoted as $h_j(\mathbf{X}; \Theta_j)$, where $\Theta_j$ parameterizes a random collection of a set of random variables of $\mathbf{X}$. In the sorting, a class assignment rule to every terminal (leaf) node $t$ considering a zero-one loss function gives:

$$h_j(\mathbf{X}; \Theta_j) = \underset{y \in \tilde{C}}{\text{argmax}}\, p(y|t) \tag{3}$$

where, we pick the class with the maximum posterior probability.

The random forest constructs $g$ by learning a series of base learners, $h_1(\mathbf{X}; \Theta_1), \cdots, h_J(\mathbf{X}; \Theta_J)$. These base learners are combined to give an ensembled prediction function $g$, determined by the most frequently predicted classes:

$$g(\mathbf{X}) = \underset{y \in \tilde{C}}{\text{argmax}} \sum_{j=1}^{J} I(y = h_j(\mathbf{X}; \Theta_j)) \tag{4}$$

where, $I$ is the indicator function. $I(y = h_j(\mathbf{X}; \Theta_j)) = 1$ if $y = h_j(\mathbf{X}; \Theta_j)$ and 0 otherwise. The number of trees in each random forest is fixed at ten, i.e., $J = 10$ for a balanced classification accuracy and computation cost (Supplementary Figure 13). We deliberately let the collaborators (clients) learn the most suitable random forest structure, i.e., the model parameters, by themselves, rather than fixing the parameters since each collaborator could have very different battery numbers and cathode material types. By only presetting the number of trees in the random forest, the collaborators could have enough flexibility to train the best model that suits their own data distribution. The bottom-level random forest algorithm (client model) is implemented using readily available MATLAB packages, more specifically, the TreeBagger function in the Statistics and Machine Learning Toolbox. The MATLAB version is R2022a, and the code runs on a personal computer with Intel (R) Core (TM) i5-10400 CPU @ 2.90 GHz RAM 8 GB.

### Federated machine learning

In the proposed federated machine learning framework, local random forests are first trained on each client with its own local data in a parallel fashion. Then the local client models are aggregated into a global model by means of a proper voting strategy from the local sorting results. In our work, battery class distribution across clients can be heterogeneous, which brings difficulty in aggregating the biased client models. To this end, we propose a Wasserstein distance voting method to aggregate the client models rather than the traditional majority voting. Our model aggregation method is robust to heterogeneous class distributions across clients. The core idea of the Wasserstein distance voting is to reduce the weightings of clients whose observations are similar to ones in the global model. The Wasserstein distance measure is defined as:

$$W_q(\cdot, \cdot) = \left( \inf_{\gamma \in \text{MP}} \int_{\Omega_1 \times \Omega_2} |x_1 - x_2|^q d\gamma(x_1, x_2) \right)^{\frac{1}{q}} \tag{5}$$

where $\gamma$ is a transport operator, referring to the transport of arbitrary attributes pairs, i.e., $(x_1, x_2)$, from the global feature space $\Omega_1$ to the client feature space $\Omega_2$. MP stands for a measurably preserved transport.

The Wasserstein distance voting term $\omega$ is defined as:

$$\omega_k = \alpha^{-\lambda(1-\mathcal{M}_k(W_q))} \tag{6}$$

where, $\alpha > 0$ and $\lambda > 0$ are voting hyperparameters. $\mathcal{M}_k$ is the average operator on the pairwise Wasserstein distance between feature spaces of $Client_k$ and the global feature space, i.e., the recycler.

The aggregated global model $G(x)$ can be obtained as:

$$G(x) = \underset{y \in \widetilde{C}}{\operatorname{argmax}} \sum_{k=1}^{K} \omega_k I(y = g_k(x)) \tag{7}$$

where, $K$ is the number of clients. $I$ is the indicator function. $I(y = h_j(\mathbf{X}; \Theta_j)) = 1$ if $y = h_j(\mathbf{X}; \Theta_j)$ and 0 otherwise. Finally, to maximally protect data privacy, the local votes are properly encrypted before being uploaded to the recycler. Most encryption methods, e.g., secure Hash Algorithms, shall suffice without compromising the sorting accuracy. The high-level federated machine learning framework, including the Wasserstein-distance voting and the transfer of parameters, is implemented from scratch.

## Feature importance

We use permutation importance to measure the feature importance in the client model, i.e., the random forest algorithm. The core idea of the permutation importance is to use out-of-bag data to examine the effect of feature permutation using the trained random forest model. In the first step, a prediction is made on several observations of the out-of-bag data. In the second step, the feature $\theta_m, m = 1, \cdots, m$ is randomly permutated for observations of out-of-bag data, then the modified out-of-bag data is passed down each tree in the random forest. In the first and second steps, two predictions are made by the trained random forest model, i.e., $\hat{C}$ and $\hat{C}^*$. The permutation feature importance of $\theta_m$ is defined as:

$$\mathrm{Imp}_n^m = \frac{1}{J_n} \sum_{j \in \mathcal{I}_n} I(y_n \neq \hat{C}_{n,j}^*) - \frac{1}{J_n} \sum_{j \in \mathcal{I}_n} I(y_n \neq \hat{C}_{n,j}) \tag{8}$$

where, $\mathscr{I}_n$ is the cardinality of the $n$ out-of-bag observations, $J_n$ is the number of trees in the random forest considering $n$ out-of-bag observations. The feature importance of $\mathrm{Imp}_n^m$ is averaged over all observations as a global importance in the client model. Similarly, feature importance in the federated machine learning framework is averaged over all clients.

## Evaluation metric

We use a one-vs-all prediction strategy to predict a multi-class classification problem, such that the original problem is converted into several binary classification problems. The accuracy of each binary classification sub-problems is defined as follows:

$$\text{accuracy} = \frac{\text{Number of correct predictions}}{\text{Total number of predictions}} = \frac{\text{TP} + \text{TN}}{\text{TP} + \text{FN} + \text{FP} + \text{TN}} \tag{9}$$

where, TP,FP,FN,TN refer to the number of true positive, false positive, false negative, and true negative predictions.

The prediction accuracy for the multi-class classification problem gives:

$$\text{Accuracy} = \frac{1}{\text{NC}} \sum_{i=1}^{\text{NC}} \text{accuracy}_i \tag{10}$$

where, $i$ is the class label, NC is the number of battery classes.

The accuracy could not provide an adequate model evaluation when classification samples are imbalanced, i.e., heterogeneous data distribution. Thus, the F1-score is used, whose definition is as follows:

$$\text{F1} = \frac{2 \times \text{Precision} \times \text{Recall}}{\text{Precision} + \text{Recall}} \tag{11}$$

where, $\text{Precision} = \text{TP}/(\text{TP} + \text{FP})$ and $\text{Recall} = \text{TP}/(\text{TP} + \text{FN})$.

## Reporting summary

Further information on research design is available in the Nature Portfolio Reporting Summary linked to this article.

## Data availability

The Center for Advanced Life Cycle Engineering (CALCE), Hawaii Natural Energy Institute (HNEI), University of Michigan (MICH), University of Oxford (OX), the Sandia National Laboratories (SNL) and Underwriters Laboratories–Purdue University (UL-PUR) datasets used in this study are available at www.batteryarchive.org. For the full details of the dataset and policies for data reuse, please refer to their website, respectively. Source data are provided with this paper.

## Code availability

Code for the modeling work is available from the corresponding authors upon request.

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

## Acknowledgements

This work was supported by the Shenzhen Science and Technology Program (Grant No. KQTD20170810150821146) [X.Z.], the Tsinghua Shenzhen International Graduate School Interdisciplinary Innovative Fund (JC2021006) [X.Z. and G.Z.], the Key Scientific Research Support Project of Shanxi Energy Internet Research Institute (SXEI2023A002) [X.Z.] and the Tsinghua Shenzhen International Graduate School-Shenzhen Pengrui Young Faculty Program of Shenzhen Pengrui Foundation (SZPR2023007) [G.Z.]. The first author would like to thank Xin Qin from the University of Cambridge, Zihao Zhou from the University of Oxford, Tsaijou Wu from Jinan University, Tingwei Cao, and Zixi Zhao from Tsinghua University for their helpful discussions in preparing the manuscript accessible to a broad readership. The authors would like to thank Prof. Qiang Yang from WeBank, as well as Ms. Yaxin Wang, for their constructive insights on federated learning. The authors would like to thank Dr. Yuliya Preger, from Sandia National Laboratory, co-founder of batteryarchive.org, for providing the first public-available repository for easy comparison of lithium-ion battery degradation data across institutions.

## Author contributions

S.T. conceptualized, designed, and performed the numerical experiments and prepared the first manuscript draft; H.L. discussed the experiments and prepared the response letter to the reviewers; C.S. contributed to the techno-environmental analysis by specifying the details of battery recycling; H.J., G.J., Z.H., R.G., and J.M. contributed to identifying the scientific issues of cathode sorting in battery recycling; R.M. and Y.C. reviewed and edited the first and revised manuscript draft; S.F., Y.W., and Y.S. contributed to the data curation and discussions; Y.R. contributed to machine learning experiment design in the revised manuscript and discussions; X.Z., G.Z., and H.S. conceptualized, reviewed, discussed, supervised this work and retrieved fundings.

## Competing interests

The authors declare no competing interests.
