## [Peer Review File · Nature Communications]

Collaborative and privacy-preserving retired battery sorting for profitable direct recycling via federated machine learningREVIEWER COMMENTS

Reviewer #1 (Remarks to the Author):

The paper presents an innovative approach to retired battery sorting using federated machine learning while considering data privacy concerns. With some clarifications, additional details, and the incorporation of comparative analysis, this work has the potential to significantly contribute to the field of battery recycling and collaborative machine learning. I recommend this paper for acceptance pending the suggested revisions.

1. To preprocess data in different clients and reduce the interference of "noisy" data, do the authors have good methods?
2. CNN-based methods usually have a better prediction accuracy based on a large dataset, why not use this method in this paper?
3. In the proposed federated machine learning framework, how to transfer parameters and ensure to avoid missing the received parameters data?

Reviewer #2 (Remarks to the Author):

The proposed paper raises an important topic in the context of sustainable development and the policy of product life cycle in the closed circular economy model. It has been proposed to use Federated Machine Learning to classify retired batteries (in particular cathode material sorting), assuming that prior information about historical operating conditions is known simultaneously in accordance with protecting the personal data of recyclers. However, I have the following concerns:

1. Thus, the decision trees are a commonly used approach, it is not known whether the authors developed and implemented the codes themselves, or whether they used ready-made Matlab-type packages or ready-made libraries in the Python language.

2. There is also no description of the neural network (including its structure/topology, number of layers, number of neurons, input description, output description, and network diagram). Some explanations/arguments for the selection of a neural network just as proposed should also be given.

3. Does the use of the proposed approach allow you to reduce the costs of recycling (taking into account the rescaling of the procedure and its implementation in practice) compared to the procedure of extracting natural resources? What is the main conclusion of the article? This should be clearly stated in the article, especially in the Abstract.

4. Thus, the information concerning testing the retired batteries at the current cycle, specifically, with a complete charging-discharging cycle is not the common approach in the case of general lithium-ion batteries. The lithium-ion batteries come not only from electric vehicles, thus the authors will praise their method as one that does not require knowledge of historical data on battery life, while very often this data is unknown, also user data are unknown (so it is not possible to use them in calculations anyway). This is possible only in the case of batteries coming from electric vehicles. What chances does the proposed technique have for practical application?

5. Some last references in the field of retired batteries with Machine Learning are also missing:

- Zhang, Y., Tang, Q., Zhang, Y. et al. Identifying degradation patterns of lithium ion batteries from impedance spectroscopy using machine learning. *Nat Commun* 11, 1706 (2020).

<https://doi.org/10.1038/s41467-020-15235-7>

- Lv, C., Zhou, X., Zhong, L., Yan, C., Srinivasan, M., Seh, Z. W., Liu, C., Pan, H., Li, S., Wen, Y., Yan, Q., Machine Learning: An Advanced Platform for Materials Development and State Prediction in Lithium-Ion Batteries. *Adv. Mater.* 2022, 34, 2101474.

<https://doi.org/10.1002/adma.202101474>

- Ng, M.F., Zhao, J., Yan, Q. et al. Predicting the state of charge and health of batteries using data-driven machine learning. *Nat Mach Intell* 2, 161–170 (2020). <https://doi.org/10.1038/s42256-020-0156-7>

6. And finally, the Figures are of low quality. This issue should be definitively improved.

Thus, this paper is very interesting and the results obtained are of high importance taking into account environmental-based issues. However, due to the above comments, I would recommend the article for publication, provided that the above concerns will be addressed. I recommend a Major Revision.

Response to Reviewer Comments

Title	Collaborative retired battery sorting for efficient and profitable recycling via federated machine learning
Revised title	Collaborative and privacy-preserving retired battery sorting for profitable direct recycling via federated machine learning
Authors	Shengyu Tao, Haizhou Liu, Chongbo Sun, Haocheng Ji, Guanjun Ji, Zhiyuan Han, Runhua Gao, Jun Ma, Ruifei Ma, Yuou Chen, Shiyi Fu, Yu Wang, Yaojie Sun, Yu Rong, Xuan Zhang, Guangmin Zhou, Hongbin Sun
Journal	Nature Communications
Manuscript ID	NCOMMS-23-27860

Table of Contents

Response to Reviewers.....	2
Response to Reviewer #1	2
Response to Reviewer #2	13
References.....	26

**Response to Reviewers**

**Response to Reviewer #1**

The paper presents an innovative approach to retired battery sorting using federated
machine learning while considering data privacy concerns. With some clarifications,
additional details, and the incorporation of comparative analysis, this work has the potential
to significantly contribute to the field of battery recycling and collaborative machine learning.
I recommend this paper for acceptance pending the suggested revisions.

Dear Respected Reviewer,

Thank you very much for your recognition of our paper. We are very pleased that you
find our work innovative and significant to battery recycling using the proposed federated
machine learning.

Your constructive comments, covering various aspects such as data preprocessing,
machine learning algorithms, and intermediate parameter transfer, are truly instrumental
in guiding us to clarify our research details and contemplate our research implications. In
the following of the section, we respectfully respond to your suggestions point by point.
Accordingly, we make careful revisions to the original manuscript and supplementary
information, where you could find the according revisions using “track changes” mode. We
truly hope that the responses appropriately address your justified concerns, and that the
revised manuscript lives up to your expectations of a decent *Nature Communications*
publication.

1. To preprocess data in different clients and reduce the interference of “noisy” data, do
the authors have good methods?

Thank you for your inquiry on the data preprocessing/denoising techniques of the
proposed federated machine learning framework. We deeply agree with you on their
importance as critical preparatory steps, as the data would heavily influence the quality of
the as-trained battery sorting models.

Data preprocessing is of critical challenge for collaborative battery recycling scenarios.
Such challenges stem from three main reasons: (1) Each client (collaborator) has diverse
preferences for data preprocessing. This divergence might be attributed to the knowledge
levels of the client's data experts, varying preprocessing costs, and different local model
accuracy requirements, which can be regarded as *human-induced noise*. (2) Each client
may possess multiple cathode material types. Heterogeneous battery data distributions
among cathode diversities might necessitate changing data preprocessing methodologies,
such as feature engineering, as well as parameter settings. This can be considered the
*cathode heterogeneity-induced noise*. (3) For each type of battery material within one
specific client, the battery testing conditions (including testing methodologies, testing
parameters, and measurement errors) can impact data quality. This can be considered
*measurement noise*. We would like to start with the *measurement noise* since it is common
to consider for all machine learning settings.

To align the preprocessing across different clients, the recycler distributes a unanimous
preprocessing protocol to the collaborators, such that all the distributed datasets are
standardized in the same fashion. The protocol involves three consecutive steps: denoising
(which you kindly pointed out), curve filling, and feature engineering.

(a) **Denoising.** Measurement noises arise inevitably from various sources, including
deviations in battery testing methods, testing parameters, and real-time measurements.
To maximally alleviate the impact of these noises, each collaborating manufacturer takes
the following 3 approaches in sequential order:

a1. Identify out-of-cycle missing data entries and refill them with their nearest non-
missing data entries.

a2. Identify in-cycle outlier data entries, and replace them with their nearest non-outlier
data entries. Here, we define a data entry as an “outlier” if it has a median deviation of at
least $n=3$ times the Median Absolute Deviation (MAD) of the original data vector.

a3. Perform moving median filtering on the data. Each data entry is replaced with the
median value of its $k=20$ neighboring entries, such that fluctuating noises within the course
of measurement can be minimized. Clients may adjust their neighboring size around the
recycler's recommendation for the best suit of data. Specifically, we set the neighboring
size for SNL and HNEI as 30 and 25 as an adjustment, while the others remaining 20.

(b) **Curve filling.** We obtain the voltage capacity and dQ/dV curves from the denoised

measurements. This includes interpolating the original voltage/capacity measurements
 with the linear interpolation function. Specifically, the recycler recommends the interpolated
 length as $L=1000$ points. Then the collaborator performs differentiation on the interpolated
 data to derive the incremental capacity curve of both the charging and discharging curves.

(c) **Feature engineering.** This includes the extraction of 30 statistical features (e.g.,
 quantiles and kurtosis) as listed in Figure 2. Further details on how the statistical features
 are defined and extracted can be found in Supplementary Note 2. We would also like to
 gently point out that feature engineering can also be considered an ex-post denoising
 process: by extracting only a few key statistics from the characteristic curves, the effect of
 most measurement noises would be neutralized.

Of course, in addition to the general procedures of preprocessing, the parameters
 employed for preprocessing should also be aligned across manufacturers. Table R1 below
 provides a list of shared parameters in preprocessing for the above steps.

**Table R1 Shared parameter values for data preprocessing**

Symbol	Meaning	Value
n	Minimum MAD ratio for an in-cycle data entry to be considered an outlier	3
k	Number of neighboring data entries to be averaged in median filtering	20,25,30
L	Interpolated voltage and capacity vector length in feature engineering	1000

 We deliberately retain *human-induced noise* and *cathode heterogeneity-induced noise*
 with the intention of making the model insensitive to these two types of noise. As previously
 mentioned, the source of *human-induced noise* arises from variations in parameter settings
 in feature engineering, as clients may decide on these settings based on the type of battery
 material. The source of *heterogeneity* stems from the possibility that a client may have
 batteries of multiple cathode material types, complicating the data distribution. The
 reported results demonstrate that federated battery recycling is insensitive to both human-
 induced noise (by allowing clients to set different data preprocessing parameters) and
 cathode heterogeneity-induced noise (by permitting clients to have a mix of different
 battery cathode material types). Consequently, our federated machine learning framework
 is broadly applicable to various cathode material sorting under both homogenous and
 heterogenous data scenarios even if allowing the collaborator to preprocess the raw data
 flexibly, showcasing the potential for scalable industry implementation.

We do realize, though, that the above data preprocessing details might not be well
 introduced in the original manuscript, which we deeply apologize for. To this end, we add
 the following description to Supplementary Note 1:

*“Recyclers often face difficulty identifying the type of retired battery, given poor access*
 *to the full historical operating data. However, battery recyclers need information on the type*
 *of retired batteries to decide on the design of recycling strategies. Therefore, it is necessary*
 *to standardize the retired battery data, even though the battery type and historical usages*

are diversified. Our proposed standardized process aims to test retired batteries with a
commonly accessible method and to obtain information about retired batteries. Specifically,
battery information is represented by a voltage-capacity curve and a dQ/dV curve derived
from the charging and discharging data, respectively. To obtain such curves, battery
recyclers need to encourage the collaborators to charge, and discharge collected retired
batteries for one cycle. The observed data on the charging and discharging characteristics,
after being recorded, are preprocessed according to the unanimous protocol distributed by
the recyclers to generate curves. The protocol first denoises measurement data by (a)
identifying out-of-cycle missing data entries and filling them with their nearest non-missing
data entries; (b) identifying in-cycle outlier data entries (defined as entries outside 3 times
the median-absolute deviation) and replacing them with their nearest non-outlier data
entries; and (c) performing median filtering based on 20 neighboring data (the neighboring
size for SNL and HNEI dataset is 30 and 25, respectively) entries to smooth the data. Next,
the protocol interpolates and differentiates the denoised data to yield the voltage capacity
and dQ/dV curves, with a recommended interpolation length of 1000 data points. Based
on the data standardization results, feature engineering can be conducted by extracting 30
key statistical features from the generated characteristic curves. It is assumed that
batteries are mandatorily decommissioned when they reach a given threshold, i.e., 80% of
the state of health (defined as the ratio of the current capacity to the nominal capacity) and
are recovered by recyclers. Therefore, data standardization here is equivalent to obtaining
information on the operating status of retired batteries at the end of their life with no
requirements on the historical operational data. It is noted that the decommission threshold
in this work is set as 90% of the nominal capacity of each battery due to the limited sample
size of the batteries that are deemed to be decommissioned.”

We sincerely hope that the above explanations have addressed your concerns
regarding the data preprocessing for federated machine learning.

2. CNN-based methods usually have a better prediction accuracy based on a large dataset,
why not use this method in this paper?

Thank you for your constructive comment on the selection of the machine learning
algorithm for federated learning. Briefly speaking, CNN is indeed one of the most powerful
algorithms in learning the underlying patterns of big data, but it might not be the best fit for
our federated learning pipeline, due to the presence of the statistical feature engineering
module as well as the computational cost-effectiveness. Please allow us to elaborate as
follows.

First, as presented in Figure 2, our proposed federated learning framework is preceded
by a statistical feature engineering procedure, which extracts 30 key features from each
pair of charging and discharging curves. Feature engineering is standard practice for most
data-driven battery analysis; see, for example, Literatures [R1] and [R2]. Such feature
engineering significantly changes the nature of the dataset, either in mechanism-driven or
in mechanism-agonistic ways, from gigabyte-scale (specifically, 4.17 GB) sequential data
to kilobyte-scale (50kB) tabular data. Decision trees such as random forests might be
better suited to this transformed dataset where the features are small-sized, non-sequential,
and highly heterogeneous [R3]. One can also better interpret the as-trained models to
analyze which extracted features are most fundamental to the determination of battery
cathode types in Figure 3d, because of the inherent feature-wise interpretability of random
forests. While it is still an open question to interpret the network behaviors.

Of course, one could argue that the framework can be designed without feature
engineering, where CNNs often yield more accurate models by automatically extracting
salient features from raw data. To this end, we randomly selected 5 batteries in each class
to demonstrate the raw charging/ discharging data, which is fed into the CNN model.

**Figure R1 The raw data of battery charging/ discharging curve in each class (after interpolation).**

In Figure R1, we interpolate the charging/ discharging data of each class of battery to
 1000 points for a standardized input size to CNN, respectively. Since the battery charging/
 discharging data are in sequential format, we choose a 1D convolutional layer as input.
 The detailed Python CNN implementation prototype is shown in Figure R2:

```
def create_cnn_model(input_shape, num_classes):
    model = Sequential()
    model.add(Conv1D(32, 3, activation='relu', kernel_initializer='he_normal', input_shape=input_shape))
    model.add(MaxPooling1D(2))
    model.add(Dropout(0.25))
    model.add(Conv1D(64, 3, activation='relu', kernel_initializer='he_normal'))
    model.add(MaxPooling1D(2))
    model.add(Dropout(0.25))
    model.add(Flatten())
    model.add(Dense(128, activation='relu', kernel_initializer='he_normal'))
    model.add(Dropout(0.5))
    model.add(Dense(num_classes, activation='softmax'))

    optimizer = tf.keras.optimizers.SGD(learning_rate=0.01, momentum=0.9)
    model.compile(optimizer=optimizer, loss='sparse_categorical_crossentropy', metrics=['accuracy'])

    return model
```

Figure R2 The implemented CNN prototype with Python3.9.13 version.

In Figure R2, we set the loss function as the sparse categorical cross-entropy loss due
 to the multi-class classification nature. We use the stochastic gradient descent, with a
 learning rate of 0.01 and momentum of 0.9, to optimize the defined loss function of the
 learning process. Given the epochs being 50 and batch size being 128, the constructed
 CNN also provides good classification when the input client data is identical to those
 producing the result in Figure 3a under the independent learning (IL) mode. Frankly,
 through case studies, we find this to be true: without feature engineering, CNN models lead
 to a comparable accuracy of 97% in Figure R3, as compared to 95% for random forests.

Figure R3 The confusion matrix of predicted and actual cathode types with client results merged.

However, such a strategy is not feasible in terms of cost-effectiveness: the construction
and training of a CNN network are computationally expensive, indicating more stringent
requirements on the data storage memory and computational power of the collaborators.
Further considering that federated learning is essentially a cross-entity iterative learning
process, CNNs might significantly slow down the training process. Based on our case study,
the model training time is recorded from each client. In Figure R4, we calculated the slope
of the least square regression line of the time-size pairs, where the value of 0.1084
indicates that one could cost *0.1084 seconds* per sample when using the as-trained CNN
model. On the contrary, the cost per sample is only *0.0008 seconds* using the random
forest reported in our work. From our perspective, using random forest would be a more
acceptable alternative for the collaborators considering the time efficiency. One may notice
the epochs are impacting the overall training time, but we find that the epochs are small
enough to produce comparable accuracy compared with the random forest. Therefore,
such a huge difference in time efficiency lands on the inherited suitability of the random
forest to the tabular data, thanks to our successful feature engineering, rather than model
parameter settings or network structure designs.

**Figure R4 The CNN training time for each client (each client has a unique sample size).**

However, thanks to your reminder, we realize that the unique advantage of random
forests is not well justified in the current manuscript. To this end, we add the following
explanations to the Discussions section:

*“Specifically, the selection of random forests as the bottom-level machine learning*
*algorithm, instead of more advanced neural network architectures, is made with full*
*consideration of the feature engineering settings and cost-effectiveness requirements.*

*Feature engineering, which prepares the data for federated learning with expert-*
*knowledge-based information extraction, transforms the raw gigabyte-scale sequential*
*data into kilobyte-scale tabular data. Decision trees such as random forests are more adept*
*at learning from such low-dimensional data with heterogeneous features, whether in terms*
*of accuracy, efficiency, or interpretability. Also, advanced neural network architectures*
*such as Convolutional Neural Networks (CNNs) require much higher computational power*
*from every collaborating manufacturer, with a significantly lengthened training time.”*

3. In the proposed federated machine learning framework, how to transfer parameters and
ensure to avoid missing the received parameters data?

Thank you very much for discussing with us the technical details of the parameter
transfer process, which is truly one of the most fundamental building blocks of federated
machine learning in real applications. In our proposal, the shared parameters (namely, the
local cathode sorting probability result) are moderately encrypted by adding random noise
at the model training stage before being directly transferred from the collaborators to the
recyclers. However, just as you kindly pointed out, the transfer process can still be fragile,
and the parameters can end up missing. Extra measures therefore need to be incorporated
to prevent or handle such undesired situations.

From our experience, the failure to receive parameters can mainly be attributed to two
causes: (a) inadvertent parameter losses due to communication delay, and computational
overload and (b) intentional parameter abduction/pollution from malicious third parties. Due
to the distinct nature of the two parameter-missing scenarios, we address them separately
as follows.

In case of inadvertent parameter losses, the most straightforward and ideal method is
to improve the stability of the distributed communicational channel and computational
resources at the hardware wireless level, such as over-the-air digital aggregation that takes
into account channel noises and perturbations [R4] [R5]. Of course, one must admit that
such random losses are inevitable at times in practical deployment, even if the systems
have been fully upgraded. Fortunately, we find that the proposed framework, consistent
with other state-of-the-art federated learning frameworks [R6] [R7], turns out to be robust
against parameter losses: even if parameters from a few manufacturers end up missing,
the subsequent Wasserstein-distance voting can still be implemented by temporarily
neglecting their presence, and the classification accuracy is only slightly degraded. This
robustness is a natural solution against inevitable parameter losses.

To prove the robustness of our WDV method, we designed a simulation experiment to
deliberately discard the parameter to be transferred. Specifically, we randomly discard the
votes from each client with a fixed percentage ratio, i.e., from 0% to 10%, and study the
response to the sorting accuracies. Once a specific vote is hacked, we set the sorting
probability of the vote to 0.5, a random guess. As illustrated in Figure R5, the WDV method
is still robust in a wide range of missing data ratios. For instance, given the heterogeneity
index is 9, the overall classification accuracy is still close to 90% under a 10% data missing
ratio, which can be considered as a severe data missing scenario. One could also find that
the result is still robust under even lower heterogeneity index settings. It is worth noticing
that the sorting accuracy decay in the horizontal direction results from the increased data
distribution complexity among clients (i.e., decreased heterogeneity index), rather than the
sensitivity to the potential data missing in the parameter transfer link.

 **Figure R5 The overall classification accuracy of the proposed WDV method under unrecoverable**
 **parameter loss and different heterogeneity settings. The client data distribution is randomly**
 **generated with the same method described in the Client Simulation Section.**

In case of malicious parameter attacks, related federated machine learning literature
 mostly chooses to design transfer mechanisms that can verify [R8] and protect [R9] the
 transferred parameters, such that third parties have fewer motivations or measures in
 maliciously maneuvering the transferred parameters. Verification approaches include
 cryptographic signatures [R10] and smart contracts in blockchains [R11]; protection
 approaches are mostly based on encryption techniques such as homomorphic encryption
 [R12] and differential privacy [R13]. In our proposed framework, thanks to your reminder,
 we have newly incorporated an RSA-based encryption scheme [R14] for both the
 verification and protection of parameters. In the verification stage, RSA is employed to
 create an avenue for verifying the authenticity of the parameter publisher. In the protection
 stage, RSA is employed to encrypt the parameters into ciphertexts. Additional technical
 approaches such as IP whitelisting/blacklisting can also be helpful in preventing third
 parties from even knowing that the parameter transfer is underway.

Despite the safe and reliable communication issues that are still open questions in
 federated machine learning, the inherent nature of feature engineering and random forests
 guarantee that there is scarcely an overwhelming number of parameters to transfer
 (kilobytes at best). This can be extremely advantageous in inhibiting parameter losses: the
 issue of inadvertent communicational and computational delays would be mild, and the
 verification/protection of parameters would not be time-consuming.

The above methods newly adopted in the framework are also introduced in the
 *Discussion* section of the manuscript as follows:

*“Supplementary Figure 10 shows that, despite such an idea assumption, the random*
 *forest model, incorporated with the Wasserstein distance voting, is naturally robust against*

*random parameter transfer losses even if parameters from a few collaborators end up*
*missing. The sorting accuracy only slightly degrades given the same heterogeneity setting.”*

We sincerely hope that you would also consider these methods suitable for addressing
different types of parameter losses.

**Response to Reviewer #2**

The proposed paper raises an important topic in the context of sustainable development
and the policy of product life cycle in the closed circular economy model. It has been
proposed to use Federal Machine Learning to classify retired batteries (in particular
cathode material sorting), assuming that prior information about historical operating
conditions is known simultaneously in accordance with protecting the personal data of
recyclers. However, I have the following concerns:

Dear Respected Reviewer,

Thank you very much for taking your precious time to review our paper. We genuinely
appreciate the positive attitude you hold towards our research paper, as well as your
constructive comments that help us further improve it.

In the following, we will address your comments in a point-by-point manner, with hopes
of further clarifying the research scope and implementation details. Accordingly, the
original manuscript and supplementary files are carefully revised, with changes highlighted
in blue in this response letter, to help readers better understand our research. Accordingly,
we made careful revisions to the original manuscript and supplementary information, where
you might find the corresponding revisions using “track changes” mode. We believe that
your professional suggestions have guided us to bring our revised manuscript up to a new
level, and hopefully, this revised version will also meet your expectations of a qualified
research paper as a decent *Nature Communications* publication.

1. Thus, the decision trees are a commonly used approach, it is not known whether the
authors developed and implemented the codes themselves, or whether they used ready-
made Matlab-type packages or ready-made libraries in the Python language.

Thank you very much for bringing into question the source of coding, which admittedly
was not well elucidated in the original manuscript. We would like to apologize for neglecting
this issue. To make up for this, we wish to make the following clarifications:

a) Random forest, as the bottom-level machine learning algorithm (i.e., the base learner
in the federated machine learning framework), is implemented with readily available
MATLAB packages (more specifically, the *TreeBagger* function in the Statistics and
Machine Learning Toolbox). Such packages enjoy high prediction performances in terms
of classification accuracy and computational efficiency and thus relieve us from the burden
of developing our own codes from scratch.

b) The high-level federated learning framework that coordinates the collaborative
learning process, specially invented Wasserstein-distance voting and the transfer of
intermediate model parameters, is coded from scratch on our own design, which is also in
MATLAB language.

We would like to open our key coding and the readers can refer to the *Code Availability*
section to see how the existing random forest packages and the self-developed federated
learning programs interact with each other to achieve the final collaborative classification.
We have also added the following explicit clarifications to the *Method-Client model and*
*Federated learning* section:

*“The bottom-level random forest algorithm (client model) is implemented using readily*
*available MATLAB packages (more specifically, the TreeBagger function in the Statistics*
*and Machine Learning Toolbox). The MATLAB version is R2022a, and the code runs on a*
*personal computer with Intel (R) Core (TM) i5-10400 CPU @ 2.90GHz RAM 8 GB.”*

...

*“The higher-level federated learning framework, including the Wasserstein-distance*
*voting and the transfer of parameters, is implemented from scratch.”*

We sincerely hope that the above clarification is sufficient for readers to understand the
detailed origins of the codes.

2. There is also no description of the neural network (including its structure/topology,
number of layers, number of neurons, input description, output description, and network
diagram). Some explanations/arguments for the selection of a neural network just as
proposed should also be given.

Thank you for kindly reminding us of the fact that we failed to provide selected values
of hyperparameters, alongside the justification for selecting these values. We are very
sorry for this neglect, as it may affect the reproducibility of codes and the evaluation of
modeling results. Moreover, we apologize for any potential confusion that we used a
neural network as the client model. Instead, we used the random forest as the client model.
Regarding the detailed description of the random forest, we only compulsorily set the
number of trees in each random forest as 10, which can be regarded as a standardized
procedure initialized by the battery recycler. In Figure R6, we use the out-of-bag
classification error to rationalize the selection of 10 as the standard number of trees in the
random forest. Note that each tree in the random forest is independently grown on a
drawn bootstrap replication of the input data. Those samples that are not in such
replication are called out-of-bag. Therefore, the out-of-bag classification error evaluates
the generalization of the unseen dataset. It turns out that the out-of-bag classification error
first rapidly decreases and then asymptotically decreases at the point where the number
of trees is 10. Since the input data of Figure R6 is from client 2, who owns up to 8 classes
of batteries, a most challenging classification task among all the heterogeneous settings
described in Supplementary Table 5, we select the number of trees as 10 to ensure an
adequate model capacity for other clients while avoiding redundant computation burden.

**Figure R6 The out-of-bag classification error against the number of trees in the random forest.**

Regarding other hyperparameters, we alternatively let the collaborators learn the most
suitable random forest structure by themselves, rather than fix the parameters since each
collaborator could have very different battery numbers and cathode material types, hence

different model parameter settings. By only presetting the number of trees in the random
 forest, the collaborators could have enough flexibility to train the best model that suits
 their own data distribution. The detailed selection of hyperparameter, i.e., the number of
 trees n , is listed in Table R2.

**Table R2 Selected hyperparameter values of the random forest algorithm**

Symbol	Meaning	Value
n	Number of trees in the random forest	10

 The model input size d_{input} and output size d_{output} are summarized in Table R3,
 where m is the sample size in each client and 30 is the extracted feature number.

**Table R3 The input and output shape of the random forest algorithm**

Symbol	Meaning	Value
d_{input}	The input size to the random forest	m by 30
d_{output}	The output size of the random forest	m by 1

 The above explanations have been added to the Methods-Client model section as
 follows:

*“The number of trees in each random forest is fixed at ten, i.e., $J = 10$ for a balanced*
 *classification accuracy and computation cost. We deliberately let the collaborators (clients)*
 *learn the most suitable random forest structure, i.e., the model parameters, by themselves,*
 *rather than fixing the parameters since each collaborator could have very different battery*
 *numbers and cathode material types. By only presetting the number of trees in the random*
 *forest, the collaborators could have enough flexibility to train the best model that suits their*
 *own data distribution. The bottom-level random forest algorithm (client model) is*
 *implemented using readily available MATLAB packages, more specifically, the TreeBagger*
 *function in the Statistics and Machine Learning Toolbox. The MATLAB version is R2022a,*
 *and the code runs on a personal computer with Intel (R) Core (TM) i5-10400 CPU @*
 *2.90GHz RAM 8 GB.”*

Here are the explanations/arguments for the selection of the random forest, rather
 than other advanced machine learning approaches, for instance, convolutional neural
 network (CNN).

Briefly speaking, advanced machine learning approaches such as CNN are indeed one
 of the most powerful algorithms in learning the underlying patterns of big data, but they
 might not be the best fit for our federated learning pipeline, due to the presence of the
 statistical feature engineering module as well as the computational cost-effectiveness.

First, as presented in Figure 2, our proposed federated learning framework is preceded

by a statistical feature engineering procedure, which extracts 30 key features from each
pair of charging/discharging curves. Feature engineering is standard practice for most
data-driven battery analysis; see, for example, Literatures [R1] and [R2]. Such feature
engineering significantly changes the nature of the dataset, from gigabyte-scale
(specifically, 4.17 GB) sequential data to kilobyte-scale (50kB) tabular data. Decision trees
such as random forests might be better suited to this transformed dataset where the
features are small-sized, non-sequential, and highly heterogeneous [R3]. One can also
better interpret the as-trained models to analyze which extracted features are most
fundamental to the determination of battery cathode types in Figure 3d, because of the
inherent feature-wise interpretability of random forests. While it is still an open question to
interpret the network behaviors.

To better clarify the motivation for selecting random forest as the base learner, we add
the following explanations to the Discussions section:

*“Specifically, the selection of random forests as the bottom-level machine learning*
*algorithm, instead of more advanced neural network architectures, is made with full*
*consideration of the feature engineering settings and cost-effectiveness requirements.*
*Feature engineering, which prepares the data for federated learning with expert-*
*knowledge-based information extraction, transforms the raw gigabyte-scale sequential*
*data into kilobyte-scale tabular data. Decision trees such as random forests are more adept*
*at learning from such low-dimensional data with heterogeneous features, whether in terms*
*of accuracy, efficiency, or interpretability. Also, advanced neural network architectures*
*such as Convolutional Neural Networks (CNNs) require much higher computational power*
*from every collaborating manufacturer, with a significantly lengthened training time and*
*compromised model interpretability.”*

3. Does the use of the proposed approach allow you to reduce the costs of recycling (taking
into account the rescaling of the procedure and its implementation in practice) compared
to the procedure of extracting natural resources? What is the main conclusion of the article?
This should be clearly stated in the article, especially in the Abstract.

We appreciate your insightful comment. In fact, using federated learning to sort cathode
materials for retired batteries cannot reduce the cost of direct battery recycling. On the
contrary, since direct battery recycling requires more cost inputs of raw materials and
chemical reagents in the pretreatment stage, its cost is higher than the procedure of
extracting natural resources (specifically, hydrometallurgy and pyrometallurgy). It is worth
noting that the excess cost of direct battery recycling is not caused by the sorting of retired
batteries assisted by federated learning. This is the intrinsic feature of the direct recycling
method that the recyclers carefully preprocess retired batteries and repair possible
electrode defects. The recycling product is functionalized electrodes or electrode materials
rather than alloy powder produced by hydrometallurgy and pyrometallurgy. The cost
comparison between direct recycling and the procedure of extracting natural resources is
shown item-wise in Figure 5a. For the same cathode material, direct recycling has the
highest cost, and the main source of excess cost is chemical reagents and raw materials
such as lithium supplements.

Pyrometallurgy recycling ultimately produces a metal alloy, while hydrometallurgy
recycling generates lithium salt and precursor materials. In contrast, direct recycling leads
to the regeneration of battery materials. These products have varying degrees of added
value, leading to differences in the profitability associated with the three recycling
strategies. Hence, even though direct recycling methods may entail slightly higher costs,
they can yield significantly greater benefits than traditional recycling methods, particularly
for NCM. However, direct recycling methods, as a promising technology, are still in the
laboratory stage. Specifically, in the laboratory, materials scientists concentrate on
addressing distinct failure issues found in specific cathode materials. Repair strategies are
highly variable, primarily due to the differing failure mechanisms observed in various
materials. To illustrate, let's consider the antisite issue. To achieve precise targeted repair,
addressing Li/Ni antisite issues in spent NCM cathodes often necessitates an oxidation
reaction to convert Ni^{2+} back to Ni^{3+} . In contrast, for spent LFP cathodes, it is typically
necessary to create a reducing environment that encourages the return of Fe^{3+} ions, which
occupy the Li layer, to their original positions. These tailored strategies align with the
specific requirements of each cathode material. Hence, to transition direct recycling to real
industrial applications, a critical first step is the rapid sorting of retired batteries.
Considering the current state of development in direct recycling technology, it is generally

challenging to directly recycle mixed and unsorted retired batteries without proper
classification.

Therefore, the main conclusion of this article is that using only field test data, rather
than battery historical data, to sort retired batteries is a key step toward the industrial
application of direct recycling. Furthermore, the accuracy of retired battery sorting will
greatly affect the profit of direct recycling, which we have analyzed in Figure 5f. Specifically,
when battery data cannot be shared due to privacy restrictions, recyclers can only use
independent learning (IL) for modeling, where the profit of direct recycling is low. This is
because a large number of retired batteries of different cathode materials are misclassified,
resulting in the wrong chemical reagents being added, further leading to the generation of
unqualified products. Using the federated learning framework we proposed, the data
privacy of collaborators is protected, and the high sorting accuracy allows recyclers to add
appropriate chemical reagents to the identified cathode materials for repair, further
generating higher recycling profits.

To better clarify the scope of our work, we have made the modifications in the Abstract,
and Conclusion part of the manuscript:

*“Unsorted retired batteries with mixed cathode materials impede the industry*
*deployment of direct recycling due to the cathode-specific nature. Given the high*
*profitability, accurately classifying the imminent surging of retired batteries is critical for the*
*commercial use of direct recycling. However, historical operation conditions, manufacturer*
*variability, and data privacy concerns from recycling collaborators (data owners) have*
*remained major challenges. In this work, we collect an out-of-distribution dataset consisting*
*of 130 lithium-ion batteries, across 5 cathode materials from 7 manufacturers. A federated*
*machine learning framework is proposed to classify these diverse retired batteries without*
*assuming any prior information on historical operation conditions and to protect the data*
*privacy of multiple recycling collaborators. With only one cycle of the end-of-life charging*
*and discharging data tested at the recycling end, our model achieves 1% and 3% cathode*
*material sorting errors using such one cycle of field available data, rather than any historical*
*data, under both homogeneous and heterogeneous recycling circumstances, respectively,*
*thanks to our proposed Wasserstein-distance voting strategy. The economic evaluation*
*shows the relevance and necessity of our accurate retired battery sorting to a profitable*
*and sustainable recycling industry in the future. This work enlightens the possibilities of*
*leveraging the existing privacy-sensitive data from multiple collaborators to develop and*
*optimize complex decision-making procedures in a collaborative and privacy-preserving*
*manner.”*

...

“Federated machine learning is a promising route for retired battery sorting and enables
emerging battery recycling technologies, *especially direct recycling*, in their development,
practical application, and optimization. We create a retired battery sorting model using only
one cycle of end-of-life charging and discharging data *as opposed to any historical data*
while preserving the data privacy budgets of multiple *battery recycling collaborators*. In the
homogeneous setting, we obtain a 1% *cathode material sorting error*; in the heterogeneous
setting, we obtain a 3% *cathode material sorting error*, thanks to our Wasserstein-distance
voting strategy. Such a level of accuracy is achieved by (1) automatically exploring the
unique patterns in the salient features without assuming any prior knowledge of historical
operation conditions and (2) using our proposed Wasserstein-distance voting strategy to
correct heterogeneous data distribution among recycling collaborators. An economic
evaluation showcases the relevance and necessity of accurate retired battery sorting to
the profitable battery recycling industry *using direct recycling*. In general, our approach can
complement the existing first-principle-based recycling route research paradigms on actual
battery recycling practice, where retired batteries are necessary while *challenging to sort*.
Broadly speaking, our work enlightens the possibilities of leveraging existing data from
multiple data owners, *rather than time-consuming and expensive data generations, to*
*develop and optimize* complex decision-making procedures such as the battery recycling
route design in a collaborative while privacy-preserving fashion.
”

4. Thus, the information concerning testing the retired batteries at the current cycle,
specifically, with a complete charging-discharging cycle is not the common approach in the
case of general lithium-ion batteries. The lithium-ion batteries come not only from electric
vehicles, thus the authors will praise their method as one that does not require knowledge
of historical data on battery life, while very often this data is unknown, also user data are
unknown (so it is not possible to use them in calculations anyway). This is possible only in
the case of batteries coming from electric vehicles. What chances does the proposed
technique have for practical application?

We sincerely appreciate the comment since it raises a critical concern about the use
case of lithium-ion batteries. As you kindly suggested a complete charging-discharging
cycle is not common in the electric vehicle scenario, which is true. However, for other use
cases such as the battery design stage, and battery manufacturing stage (quality control),
the complete charging-discharging cycle is standard since the battery designer and the
battery manufacturer are finding valuable insights into the battery material formula, and
manufacturing variabilities. The core idea of using complete charging and discharging
cycles is to ensure that no extra variabilities are introduced by dynamic charging or
discharging. We also take a similar idea when dealing with the battery recycling scenario.
Considering the highly heterogeneous data distribution of the retired batteries, if the
recycler wants to gain insights into the diversified cathode material types by studying the
voltage response curves, then the complete charging and discharging procedures should
be standardized to avoid any extra variabilities.

We also admit that lithium-ion batteries have not only retired from electric vehicles but
also from various applications such as data center energy storage systems, power grid
energy storage systems, and consumer electronics. As you kindly commented, the battery
data is very often unknown, which is also the starting point of our idea that the battery
recycling scenario has very diversified battery origins and has no access to historical data.
Therefore, it is not possible to use the historical data in calculations anyway. To this end,
we use the data at the “current cycle” to emphasize that our method requires no historical
information on the retired batteries. Specifically, the concept of the “current cycle” means
that the battery recycler tests the retired batteries with a charging/ discharging procedure,
regardless of any historical use conditions. However, we admit that this “current cycle”
could lead to potential confusion. We have accordingly made corrections in the Abstract,
Conclusion section, and relevant phrases in the main body of the manuscript (only the
modified Abstract and Conclusion part is pasted below for your easier reference):

*“...With only one cycle of the end-of-life charging and discharging data tested at the*
*recycling end, our model achieves 1% and 3% cathode material sorting errors using such*
*one cycle of field available data, rather than any historical data...”*

...

*“...We create a retired battery sorting model using only one cycle of end-of-life*
*charging and discharging data as opposed to any historical data while preserving the data*
*privacy budgets of multiple battery recycling collaborators...Such a level of accuracy is*
*achieved by (1) automatically exploring the unique patterns in the salient features without*
*assuming any prior knowledge of historical operation conditions and...”*

Therefore, our proposed method is designed in the first place for practical applications
under the retired battery sorting scenario, where historical data access is hardly available.
Despite the diversified cathode material types of the retired batteries, the recycling
collaborators only charge and discharge the batteries for one cycle to retrieve the field-
testing data, which brings huge flexibility to the collaborative retired battery sorting. We
even set no restrictions on the origin of the retired batteries, such that the trained model is
more generalized in practical battery recycling scenarios. We hope that our explanation
could alleviate your justified concerns about the practical issues in the industrialization of
direct battery recycling.

5. Some last references in the field of retired batteries with machine learning are also
missing:

- Zhang, Y., Tang, Q., Zhang, Y. et al. Identifying degradation patterns of lithium ion
batteries from impedance spectroscopy using machine learning. Nat Commun 11, 1706
(2020). <https://doi.org/10.1038/s41467-020-15235-7>

- Lv, C., Zhou, X., Zhong, L., Yan, C., Srinivasan, M., Seh, Z. W., Liu, C., Pan, H., Li, S.,
Wen, Y., Yan, Q., Machine Learning: An Advanced Platform for Materials Development
and State Prediction in Lithium-Ion Batteries. Adv. Mater. 2022, 34, 2101474.
<https://doi.org/10.1002/adma.202101474>

- Ng, MF., Zhao, J., Yan, Q. et al. Predicting the state of charge and health of batteries
using data-driven machine learning. Nat Mach Intell 2, 161–170 (2020).
<https://doi.org/10.1038/s42256-020-0156-7>

Thank you very much for your suggestion on the list of references. After a careful read
of all three papers, we do find them to be highly relevant to our paper, as they are all
centered around the topic of machine-learning-based battery performance evaluation.
Moreover, they focus on different stages of the battery life cycle, including development,
in-service, and recycling, and are all highly accredited in their respective areas. Citing the
mentioned literature would significantly increase the comprehensiveness of the existing
literature review. Therefore, in the revised manuscript, we include these references in the
*Introduction* section:

*“In other battery-related topics, machine learning has recently allowed us to*
*automatically discover complex battery mechanisms [17-19], predict remaining useful life*
*[20-23], evaluate the state of health [19, 24, 25], optimize the cycling profile [26, 27], and*
*approximate the failure distribution [28], even to guide the battery design [29, 30] and*
*predict life-long performance immediately after manufacturing [31]...*

*[19] Zhang Y, Tang Q, Zhang Y, Wang J, Stimming U, Lee A A. Identifying degradation*
*patterns of lithium ion batteries from impedance spectroscopy using machine learning.*
*Nature Communications 11. 2020.*

*[25] Ng M-F, Zhao J, Yan Q, Conduit G J, Seh Z W. Predicting the state of charge and*
*health of batteries using data-driven machine learning. Nature Machine Intelligence 2.*
*2020:161-70.*

*[30] Lv C, Zhou X, Zhong L, Yan C, Srinivasan M, Seh Z W, et al. Machine learning:*
*an advanced platform for materials development and state prediction in lithium-ion*
*batteries. Advanced Materials 34(25). 2022: 2101474.”*

Again, we would like to express our gratitude to you for recommending such high-
quality journals as references, so that the literature review can now be complete and more
well-rounded.

6. And finally, the figures are of low quality. This issue should be definitively improved.

Thank you very much for pointing out the quality issue of the figures. We are truly sorry
that we failed to generate the figures in a clear and readable format, which must have
caused you many unnecessary troubles in trying to review our work.

To fix this issue, we have regenerated all figures with higher resolution, and have made
sure that the figure quality is not degraded when the manuscript is converted to PDF format.
Additionally, we enlarge the fonts for most figures in the manuscript and supplementary
information, so that readers can read information more conveniently.

We sincerely hope that you will also find the revised figures decent in quality and ready
for publication.

Thus, this paper is very interesting and the results obtained are of high importance taking
into account environmental-based issues. However, due to the above comments, I would
recommend the article for publication, provided that the above concerns will be addressed.
I recommend a Major Revision.

Again, we would like to thank you so much for agreeing to review our paper, and for
providing such positive and constructive feedback. We sincerely hope that our carefully
prepared responses and revisions can alleviate all your concerns about our proposed
federated battery classification method and that the paper is more suitable for publication
in *Nature Communications*.

**References**

- [R1] Zhang Y, Tang Q, Zhang Y, Wang J, Stimming U, Lee AA. Identifying degradation patterns of lithium
ion batteries from impedance spectroscopy using machine learning. *Nature communications*. 2020
Apr 6;11(1):1706.
- [R2] Roman D, Saxena S, Robu V, Pecht M, Flynn D. Machine learning pipeline for battery state-of-health
estimation. *Nature Machine Intelligence*. 2021 May;3(5):447-56.
- [R3] Shwartz-Ziv R, Armon A. Tabular data: Deep learning is not all you need. *Information Fusion*. 2022
May 1;81:84-90.
- [R4] Zhu G, Du Y, Gündüz D, Huang K. One-bit over-the-air aggregation for communication-efficient
federated edge learning: Design and convergence analysis. *IEEE Transactions on Wireless
Communications*. 2020 Nov 26;20(3):2120-35.
- [R5] Elgabri A, Park J, Issaid CB, Bennis M. Harnessing wireless channels for scalable and privacy-
preserving federated learning. *IEEE Transactions on Communications*. 2021 May 10;69(8):5194-
208.
- [R6] Ye H, Liang L, Li GY. Decentralized federated learning with unreliable communications. *IEEE journal
of selected topics in signal processing*. 2022 Feb 18;16(3):487-500.
- [R7] Han Z, Zhou L, Ge C, Li J, Liu Z. Robust privacy - preserving federated learning framework for IoT
devices. *International Journal of Intelligent Systems*. 2022 Nov;37(11):9655-73.
- [R8] Peng Z, Xu J, Chu X, Gao S, Yao Y, Gu R, Tang Y. Vfchain: Enabling verifiable and auditable
federated learning via blockchain systems. *IEEE Transactions on Network Science and Engineering*.
2021 Jan 12;9(1):173-86.
- [R9] Qi T, Wu F, Wu C, He L, Huang Y, Xie X. Differentially private knowledge transfer for federated
learning. *Nature Communications*. 2023 Jun 24;14(1):3785.
- [R10] Kanchan S, Choi BJ. Group Signature Based Federated Learning Approach for Privacy Preservation.
In *2021 International Conference on Electrical, Computer and Energy Technologies (ICECET) 2021
Dec 9 (pp. 1-6)*. IEEE.
- [R11] Kim H, Park J, Bennis M, Kim SL. Blockchain-based on-device federated learning. *IEEE Communications
Letters*. 2019 Jun 10;24(6):1279-83.
- [R12] Fang H, Qian Q. Privacy preserving machine learning with homomorphic encryption and federated
learning. *Future Internet*. 2021 Apr 8;13(4):94.
- [R13] Wei K, Li J, Ding M, Ma C, Yang HH, Farokhi F, Jin S, Quek TQ, Poor HV. Federated learning with
differential privacy: Algorithms and performance analysis. *IEEE Transactions on Information
Forensics and Security*. 2020 Apr 17;15:3454-69.
- [R14] Zhou X, Tang X. Research and implementation of RSA algorithm for encryption and decryption. In
*Proceedings of 2011 6th international forum on strategic technology 2011 Aug 22 (Vol. 2, pp. 1118-
1121)*. IEEE.

REVIEWERS' COMMENTS

Reviewer #1 (Remarks to the Author):

I have carefully reviewed the revised manuscript, and I appreciate the effort put into addressing the concerns raised during the initial review. The authors have made significant improvements to the paper, and many of the previously identified issues have been adequately addressed. Based on my assessment, I recommend accepting this manuscript for publication.

Reviewer #2 (Remarks to the Author):

The Authors took my comments into account and I recommend the paper for publication.

Response to Reviewer Comments

Title	Collaborative and privacy-preserving retired battery sorting for profitable direct recycling via federated machine learning
Authors	Shengyu Tao, Haizhou Liu, Chongbo Sun, Haocheng Ji, Guanjun Ji, Zhiyuan Han, Runhua Gao, Jun Ma, Ruifei Ma, Yuou Chen, Shiyi Fu, Yu Wang, Yaojie Sun, Yu Rong, Xuan Zhang, Guangmin Zhou, Hongbin Sun
Journal	Nature Communications
Manuscript ID	NCOMMS-23-27860A

Table of Contents

Response to Reviewers.....	2
Response to Reviewer #1	2
Response to Reviewer #2.....	3

**Response to Reviewers**

**Response to Reviewer #1**

I have carefully reviewed the revised manuscript, and I appreciate the effort put into
addressing the concerns raised during the initial review. The authors have made significant
improvements to the paper, and many of the previously identified issues have been
adequately addressed.

Based on my assessment, I recommend accepting this manuscript for publication.

Dear Respected Reviewer,

We express our sincere thanks to your time and professional review comments. We
are very pleased that you find our work lives up to your expectations of a decent *Nature*
*Communications* publication.

**Response to Reviewer #2**

The Authors took my comments into account and I recommend the paper for publication.

Dear Respected Reviewer,

Thank you very much for taking your precious time to review our paper. We genuinely
appreciate the positive attitude you hold towards our research paper, which meets your
expectations of a qualified research paper as a decent *Nature Communications* publication.